# Certifying Some Distributional Fairness with Subpopulation Decomposition

**Mintong Kang** [*]
UIUC
`mintong2@illinois.edu`

**Linyi Li** [*]
UIUC
`linyi2@illinois.edu`

**Maurice Weber**
ETH Zurich
`maurice.weber@inf.ethz.ch`

**Yang Liu**
UC Santa Cruz
`yangliu@ucsc.edu`

**Ce Zhang**
ETH Zurich
`ce.zhang@inf.ethz.ch`

**Bo Li**
UIUC
`lbo@illinois.edu`

## Abstract

Extensive efforts have been made to understand and improve the fairness of machine learning models based on different fairness measurement metrics, especially in high-stakes domains such as medical insurance, education, and hiring decisions. However, there is a lack of *certified fairness* on the end-to-end performance of an ML model. In this paper, we first formulate the certified fairness of an ML model trained on a given data distribution as an optimization problem based on the model performance loss bound on a fairness constrained distribution, which is within bounded distributional distance with the training distribution. We then propose a general fairness certification framework and instantiate it for both sensitive shifting and general shifting scenarios. In particular, we propose to solve the optimization problem by decomposing the original data distribution into analytical subpopulations and proving the convexity of the sub-problems to solve them. We evaluate our certified fairness on six real-world datasets and show that our certification is tight in the sensitive shifting scenario and provides non-trivial certification under general shifting. Our framework is flexible to integrate additional non-skewness constraints and we show that it provides even tighter certification under different real-world scenarios. We also compare our certified fairness bound with adapted existing distributional robustness bounds on Gaussian data and demonstrate that our method is significantly tighter.

## 1 Introduction

As machine learning (ML) has become ubiquitous [24, 18, 5, 11, 8, 13], fairness of ML have attracted a lot of attention from different perspectives. For instance, some automated hiring systems are biased towards males due to gender imbalanced training data [3]. Different approaches have been proposed to improve ML fairness, such as regularized training [16, 22, 26, 30], disentanglement [12, 28, 40], duality [44], low-rank matrix factorization [34], and distribution alignment [4, 29, 53].

In addition to existing approaches that *evaluate* fairness, it is important and challenging to provide *certification* for ML fairness. Recent studies have explored the certified fair *representation* of ML [39, 4, 36]. However, there lacks certified fairness on the *predictions* of an end-to-end ML model trained on an arbitrary data distribution. In addition, current fairness literature mainly focuses on training an ML model on a potentially (im)balanced distribution and evaluate its performance in a target domain measured by existing statistical fairness definitions [17, 20]. Since in practice these selected target domains can encode certain forms of unfairness of their own (e.g., sampling bias), the evaluation would be more informative if we can evaluate and certify fairness of an ML model

---

[*]The first two authors contributed equally.

on an *objective* distribution. Taking these factors into account, in this work, we aim to provide the first definition of *certified fairness* given an ML model and a training distribution by bounding its end-to-end performance on an objective, *fairness constrained distribution*. In particular, we define *certified fairness* as the worst-case upper bound of the ML prediction loss on a fairness constrained test distribution $\mathcal{Q}$, which is within a bounded distance to the training distribution $\mathcal{P}$. For example, for an ML model of crime rate prediction, we can define the model performance as the expected loss within a specific age group. Suppose the model is deployed in a fair environment that does not deviate too much from the training, our fairness certificate can guarantee that the loss of crime rate prediction for a particular age group is upper bounded, which is an indicator of model's fairness.

We mainly focus on the base rate condition as the fairness constraint for $\mathcal{Q}$. We prove that our certified fairness based on a base rate constrained distribution will imply other fairness metrics, such as demographic parity (DP) and equalized odds (EO). Moreover, our framework is flexible to integrate other fairness constraints into $\mathcal{Q}$. We consider two scenarios: (1) *sensitive shifting* where only the joint distribution of sensitive attribute and label can be changed when optimizing $\mathcal{Q}$; and (2) *general shifting* where everything including the conditioned distribution of non-sensitive attributes can be changed. We then propose an effective *fairness certification framework* to compute the certificate.

In our fairness certification framework, we first formulate the problem as constrained optimization, where the fairness constrained distribution is encoded by base rate constraints. Our key technique is to decompose both training and the fairness constrained test distributions to several subpopulations based on sensitive attributes and target labels, which can be used to encode the base rate constraints. With such a decomposition, in sensitive shifting, we can decompose the distance constraint to subpopulation ratio constraints and prove the transformed low-dimensional optimization problem is convex and thus efficiently solvable. In general shifting case, we propose to solve it based on divide and conquer: we first partition the feasible space into different subpopulations, then optimize the density (ratio) of each subpopulation, apply relaxation on each subpopulation as a sub-problem, and finally prove the convexity of the sub-problems with respect to other low-dimensional variables. Our framework is applicable for any black-box ML models and any distributional shifts bounded by the Hellinger distance, which is a type of $f$-divergence studied in the literature [47, 14, 7, 25, 15].

To demonstrate the effectiveness and tightness of our framework, we evaluate our fairness bounds on *six* real-world fairness related datasets [3, 2, 19, 48]. We show that our certificate is tight under different scenarios. In addition, we verify that our framework is flexible to integrate additional constraints on $\mathcal{Q}$ and evaluate the certified fairness with additional non-skewness constraints, with which our fairness certificate is tighter. Finally, as the first work on certifying fairness of an end-to-end ML model, we adapt existing distributional robustness bound [43] for comparison to provide more intuition. Note that directly integrating the fairness constraint to the existing distributional robustness bound is challenging, which is one of the main contributions for our framework. We show that with the fairness constraints and our effective solution, our bound is strictly tighter.

**Technical Contributions**. In this work, we take the first attempt towards formulating and computing the *certified fairness* on an end-to-end ML model, which is trained on a given distribution. We make contributions on both theoretical and empirical fronts.

1. We formulate the *certified fairness* of an end-to-end ML model trained on a given distribution $\mathcal{P}$ as the worst-case upper bound of its prediction loss on a fairness constrained distribution $\mathcal{Q}$, which is within bounded distributional distance with $\mathcal{P}$.

2. We propose an effective fairness certification framework that simulates the problem as constrained optimization and solve it by decomposing the training and fairness constrained test distributions into subpopulations and proving the convexity of each sub-problem to solve it.

3. We evaluate our certified fairness on six real-world datasets to show its tightness and scalability. We also show that with additional distribution constraints on $\mathcal{Q}$, our certification would be tighter.

4. We show that our bound is strictly tighter than adapted distributional robustness bound on Gaussian dataset due to the added fairness constraints and our effective optimization approach.

**Related Work**   Fairness in ML can be generally categorized into individual fairness and group fairness. Individual fairness guarantees that similar inputs should lead to similar outputs for a model and it is analyzed with optimization approaches [49, 33] and different types of relaxations [21]. Group fairness indicates to measure the *independence* between the sensitive features and model prediction, the *separation* which means that the sensitive features are statistically independent of model prediction given the target label, and the *sufficiency* which means that the sensitive features are

statistically independent of the target label given the model prediction [27]. Different approaches are proposed to analyze group fairness via static analysis [46], interactive computation [41], and probabilistic approaches [1, 10, 6]. In addition, there is a line of work trying to certify the *fair representation* [39, 4, 36]. In [9], the authors have provided bounds for how group fairness transfers subject to bounded distribution shift. Our certified fairness differs from existing work from three perspectives: 1) we provide fairness certification considering the end-to-end model performance instead of the representation level, 2) we define and certify fairness based on a fairness constrained distribution which implies other fairness notions, and 3) our certified fairness can be computed for *any* black-box models trained on an arbitrary given data distribution.

## 2 Certified Fairness Based on Fairness Constrained Distribution

In this section, we first introduce preliminaries, and then propose the definition of *certified fairness* based on a bounded fairness constrained distribution, which to the best of our knowledge is the first formal fairness certification on end-to-end model prediction. We also show that our proposed certified fairness relates to established fairness definitions in the literature.

**Notations.** We consider the general classification setting: we denote by $\mathcal{X}$ and $\mathcal{Y} = [C]$ the feature space and labels, $[C] := \{1, 2, \cdots, C\}$. $h_\theta \colon \mathcal{X} \to \Delta^{|\mathcal{Y}|}$ represents a mapping function parameterized with $\theta \in \Theta$, and $\ell \colon \Delta^{|\mathcal{Y}|} \times \mathcal{Y} \to \mathbb{R}_+$ is a non-negative loss function such as cross-entropy loss. Within feature space $\mathcal{X}$, we identify a *sensitive* or *protected attribute* $\mathcal{X}_s$ that takes a finite number of values: $\mathcal{X}_s := [S]$, i.e., for any $X \in \mathcal{X}$, $X_s \in [S]$.

**Definition 1** (Base Rate). Given a distribution $\mathcal{P}$ supported over $\mathcal{X} \times \mathcal{Y}$, the base rate for sensitive attribute value $s \in [S]$ with respect to label $y \in [C]$ is $b_{s,y}^{\mathcal{P}} = \Pr_{(X,Y)\sim\mathcal{P}}[Y = y \mid X_s = s]$.

Given the definition of base rate, we define a *fair base rate distribution* (in short as *fair distribution*).

**Definition 2** (Fair Base Rate Distribution). A distribution $\mathcal{P}$ supported over $\mathcal{X} \times \mathcal{Y}$ is a fair base rate distribution if and only if for any label $y \in [C]$, the base rate $b_{s,y}^{\mathcal{P}}$ is equal across all $s \in [S]$, i.e., $\forall i \in [S], \forall j \in [S], b_{i,y}^{\mathcal{P}} = b_{j,y}^{\mathcal{P}}$.

*Remark.* In the literature, the concepts of fairness are usually directly defined at the model prediction level, where the criterion is whether the model prediction is fair against individual attribute changes [39, 36, 50] or fair at population level [54]. In this work, to certify the fairness of model prediction, we define a fairness constrained distribution on which we will certify the model prediction (e.g., bound the prediction error), rather than relying on the empirical fairness evaluation. In particular, we first define the fairness constrained distribution through the lens of base rate parity, i.e., the probability of being any class should be independent of sensitive attribute values, and then define the certified fairness of a given model based on its performance on the fairness constrained distribution as we will show next.

The choice of focusing on fair base rate may look restrictive but its definition aligns very well with the celebrated fairness definition Demographic Parity [51], which promotes that $\Pr[h_\theta(X) = 1|X_s = i] = \Pr[h_\theta(X) = 1|X_s = j]$. In this case, the prediction performance of $h_\theta$ on $\mathcal{Q}$ with fair base rate will relate directly to $\Pr[h_\theta(X) = 1|X_s = i]$. Secondly, under certain popular data generation process, the base rate sufficiently encodes the differences in distributions and a fair base rate will imply a homogeneous (therefore equal or "fair") distribution over $X, Y$: consider when $\Pr(X|Y = y, X_s = i)$ is the same across different group $X_s$. Then $\Pr(X, Y|X_s = i)$ is simply a linear combination of basis distributions $\Pr(X|Y = y, X_s = i)$, and the difference between different groups' joint distribution of $X, Y$ is fully characterized by the difference in base rate $\Pr(Y = y|X_s)$. This assumption will greatly enable trackable analysis and is not an uncommon modeling choice in the recent discussion of fairness when distribution shifts [52, 37].

### 2.1 Certified Fairness

Now we are ready to define the fairness certification based on the optimized fairness constrained distribution. We define the certification under two data generation scenarios: *general shifting* and *sensitive shifting*. In particular, consider the data generative model $\Pr(X_o, X_s, Y) = \Pr(Y)\Pr(X_s|Y)\Pr(X_o|Y, X_s)$, where $X_o$ and $X_s$ represent the non-sensitive and sensitive features, respectively. If all three random variables on the RHS are allowed to change, we call it *general shifting*; if both $\Pr(Y)$ and $\Pr(X_s|Y)$ are allowed to change to ensure the fair base rate (Def. 2)

while $\Pr(X_o|Y, X_s)$ is the same across different groups, we call it *sensitive shifting*. In Section 3 we will introduce our certification framework for both scenarios.

**Problem 1** (Certified Fairness with General Shifting). *Given a training distribution $\mathcal{P}$ supported on $\mathcal{X} \times \mathcal{Y}$, a model $h_\theta(\cdot)$ trained on $\mathcal{P}$, and distribution distance bound $\rho > 0$, we call $\bar{\ell} \in \mathbb{R}$ a fairness certificate with general shifting, if $\bar{\ell}$ upper bounds*

$$\max_{\mathcal{Q}} \mathbb{E}_{(X,Y)\sim\mathcal{Q}}[\ell(h_\theta(X), Y)] \quad \text{s.t.} \quad \text{dist}(\mathcal{P}, \mathcal{Q}) \leq \rho, \quad \mathcal{Q} \text{ is a fair distribution,}$$

*where $\text{dist}(\cdot, \cdot)$ is a predetermined distribution distance metric.*

In the above definition, we define the fairness certificate as the upper bound of the model's loss among all fair base rate distributions $\mathcal{Q}$ within a bounded distance from $\mathcal{P}$. Besides the bounded distance constraint $\text{dist}(\mathcal{P}, \mathcal{Q}) \leq \rho$, there is no other constraint between $\mathcal{P}$ and $\mathcal{Q}$ so this satisfies "*general shifting*". This bounded distance constraint, parameterized by a tunable parameter $\rho$, ensures that the test distribution should not be too far away from the training. In practice, the model $h_\theta$ may represent a DNN whose complex analytical forms would pose challenges for solving Problem 1. As a result, as we will show in Equation (2) we can query some statistics of $h_\theta$ trained on $\mathcal{P}$ as constraints to characterize $h_\theta$, and thus compute the upper bound certificate.

The feasible region of optimization problem 1 might be empty if the distance bound $\rho$ is too small, and thus we cannot provide fairness certification in this scenario, indicating that there is no nearby fair distribution and thus the fairness of the model trained on the highly "unfaired" distribution is generally low. In other words, if the training distribution $\mathcal{P}$ is unfair (typical case) and there is no feasible fairness constrained distribution $\mathcal{Q}$ within a small distance to $\mathcal{P}$, fairness cannot be certified.

This definition follows the intuition of typical real-world scenarios: The real-world training dataset is usually biased due to the limitation in data curation and collection processes, which causes the model to be unfair. Thus, when the trained models are evaluated on the real-world fairness constrained test distribution or ideal fair distribution, we hope that the model does not encode the training bias which would lead to low test performance. That is to say, the model performance on fairness constrained distribution is indeed a witness of the model's intrinsic fairness.

We can further constrain that the subpopulation of $\mathcal{P}$ and $\mathcal{Q}$ parameterized by $X_s$ and $Y$ does not change, which results in the following "*sensitive shifting*" fairness certification.

**Problem 2** (Certified Fairness with Sensitive Shifting). *Under the same setting as Problem 1, we call $\bar{\ell}$ a fairness certificate against sensitive shifting, if $\bar{\ell}$ upper bounds*

$$\max_{\mathcal{Q}} \mathbb{E}_{(X,Y)\sim\mathcal{Q}}[\ell(h_\theta(X), Y)]$$
$$\text{s.t.} \quad \text{dist}(\mathcal{P}, \mathcal{Q}) \leq \rho, \quad \mathcal{P}_{s,y} = \mathcal{Q}_{s,y} \, \forall s \in [S], y \in [C], \quad \mathcal{Q} \text{ is a fair distribution,}$$

*where $\mathcal{P}_{s,y}$ and $\mathcal{Q}_{s,y}$ are the subpopulations of $\mathcal{P}$ and $\mathcal{Q}$ on the support $\{(X, Y) : X \in \mathcal{X}, X_s = s, Y = y\}$ respectively, and $\text{dist}(\cdot, \cdot)$ is a predetermined distribution distance metric.*

The definition adds an additional constraint between $\mathcal{P}$ and $\mathcal{Q}$ that each subpopulation, partitioned by the sensitive attribute $X_s$ and label $Y$, does not change. This constraint corresponds to the scenario where the distribution shifting between training and test distributions only happens on the proportions of different sensitive attributes and labels, and within each subpopulation the shifting is negligible.

In addition, to model the real-world test distribution, we may further request that the test distribution $\mathcal{Q}$ is not too skewed regarding the sensitive attribute $X_s$ by adding constraint (1). We will show that this constraint can also be integrated into our fairness certification framework flexibly in Section 4.3.

$$\forall i \in [S], \forall j \in [S], \left| \Pr_{(X,Y)\sim\mathcal{Q}}[X_s = i] - \Pr_{(X,Y)\sim\mathcal{Q}}[X_s = j] \right| \leq \Delta_S. \tag{1}$$

**Connections to Other Fairness Measurements.** Though not explicitly stated, our goal of certifying the performance on a fair distribution $\mathcal{Q}$ relates to certifying established fairness definitions in the literature. Consider the following example: Suppose Problem 2 is feasible and returns a classifier $h_\theta$ that achieves certified fairness per group and per label class $l := \Pr_{(X,Y)\sim\mathcal{Q}}[h_\theta(X) \neq Y | Y = y, X_s = i] \leq \epsilon$ on $\mathcal{Q}$. We will then have the following proposition:

**Proposition 1.** $h_\theta$ *achieves $\epsilon$-Demographic Parity (DP) [51] and $\epsilon$-Equalized Odds (EO) [18]:*

- *$\epsilon$-DP:* $|\Pr_{\mathcal{Q}}[h_\theta(X) = 1 | X_s = i] - \Pr_{\mathcal{Q}}[h_\theta(X) = 1 | X_s = j]| \leq \epsilon, \, \forall i, j.$

- *ε-EO:* $|\mathrm{Pr}_{\mathcal{Q}}[h_\theta(X) = 1|Y = y, X_s = i] - \mathrm{Pr}_{\mathcal{Q}}[h_\theta(X) = 1|Y = y, X_s = j]| \le \epsilon, \forall y, i, j.$

*Remark.* The detailed proof is omitted to appendix C.1. (1) When $\epsilon = 0$, Proposition 1 can guarantee perfect DP and EO simultaneously. We achieve so because we evaluate with a fair distribution $\mathcal{Q}$, where "fair distribution" stands for "equalized base rate" and according to [23, Theorem 1.1, page 5] both DP and EO are achievable for this fair distribution. This observation in fact motivated us to identify the fair distribution $\mathcal{Q}$ for the evaluation since it is this fair distribution that allows the fairness measures to hold at the same time. Therefore, another way to interpret our framework is: given a model, we provide a framework that certifies worst-case "unfairness" bound in the context where perfect fairness is achievable. Such a worse-case bound serves as the gap to a perfectly fair model and could be a good indicator of the model's fairness level. (2) In practice, $\epsilon$ is not necessarily zero. Therefore, Proposition 1 only provides an upper lower bound of DP and EO, namely $\epsilon$-DP and $\epsilon$-EO, instead of absolute DP and EO. The approximate fairness guarantee renders our results more general. Meanwhile, there is a higher flexiblity in simultaneously satisfying approximate fairness metrics (for example when DP = 0, but EO = $\epsilon$, which is plausible for a proper range of epsilon, regardless of the distribution $\mathcal{Q}$ being fair or not). But again, similar to (1), $\epsilon$-DP and $\epsilon$-EO can be achieved at the same time easily since the test distribution satisfies base rate parity.

The bounds in Proposition 1 are tight. Consider the distribution $\mathcal{Q}$ with binary classes and binary sensitive attributes (i.e., $Y, X_s \in \{0, 1\}$). When the distribution $\mathcal{Q}$ and classifier $h_\theta$ satisfy the conditions that $\mathrm{Pr}_{\mathcal{Q}}[h_\theta(X) \ne Y|Y = 0, X_s = 0] = \epsilon, \mathrm{Pr}_{\mathcal{Q}}[h_\theta(X) \ne Y|Y = 0, X_s = 1] = 0$ and $\mathrm{Pr}_{\mathcal{Q}}[Y = 0] = 1, \mathrm{Pr}_{\mathcal{Q}}[Y = 1] = 0$, the bounds in Proposition 1 are tight. From $\mathrm{Pr}_{\mathcal{Q}}[Y = 0] = 1, \mathrm{Pr}_{\mathcal{Q}}[Y = 1] = 0$, we can observe that $\epsilon$-DP is equivalent to $\epsilon$-EO. From $\mathrm{Pr}_{\mathcal{Q}}[h_\theta(X) \ne Y|Y = 0, X_s = 0] = \epsilon, \mathrm{Pr}_{\mathcal{Q}}[h_\theta(X) \ne Y|Y = 0, X_s = 1] = 0$ and $\mathrm{Pr}_{\mathcal{Q}}[h_\theta(X) \ne Y|Y = 0, X_s = i] = \mathrm{Pr}_{\mathcal{Q}}[h_\theta(X) = 1|Y = 0, X_s = i]$ for $i \in \{0, 1\}$, we know that $\epsilon$-EO holds with tightness since $|\mathrm{Pr}_{\mathcal{Q}}[h_\theta(X) = 1|Y = 0, X_s = 0] - \mathrm{Pr}_{\mathcal{Q}}[h_\theta(X) = 1|Y = 0, X_s = 1]| = \epsilon$. To this point, we show that both bounds in Proposition 1 are tight.

## 3 Fairness Certification Framework

We will introduce our fairness certification framework which efficiently computes the fairness certificate defined in Section 2.1. We first introduce our framework for *sensitive shifting* (Problem 2) which is less complex and shows our core methodology, then *general shifting* case (Problem 1).

Our framework focuses on using the Hellinger distance to bound the distributional distance in Problems 1 and 2. The Hellinger distance $H(\mathcal{P}, \mathcal{Q})$ is defined in Def. 3 (in Appendix B.1). The Hellinger distance has some nice properties, e.g., $H(\mathcal{P}, \mathcal{Q}) \in [0, 1]$, and $H(\mathcal{P}, \mathcal{Q}) = 0$ if and only if $\mathcal{P} = \mathcal{Q}$ and the maximum value of 1 is attained when $\mathcal{P}$ and $\mathcal{Q}$ have disjoint support. The Hellinger distance is a type of $f$-divergences which are widely studied in ML distributional robustness literature [47, 14] and in the context of distributionally robust optimization [7, 25, 15]. Also, using Hellinger distance enables our certification framework to generalize to *total variation distance (or statistic distance)* $\delta(\mathcal{P}, \mathcal{Q})$[2] directly with the connection, $H^2(\mathcal{P}, \mathcal{Q}) \le \delta(\mathcal{P}, \mathcal{Q}) \le \sqrt{2}H(\mathcal{P}, \mathcal{Q})$ ([45], Equation 1). We leave the extension of our framework to other distance metrics as future work.

### 3.1 Core Idea: Subpopulation Decomposition

The core idea in our framework is (finite) subpopulation decomposition. Consider a generic optimization problem for computing the loss upper bound on a constrained test distribution $\mathcal{Q}$, given training distribution $\mathcal{P}$ and trained model $h_\theta(\cdot)$, we first characterize model $h_\theta(\cdot)$ based on some statistics, e.g., mean and variance for loss of the model: $h_\theta(\cdot)$ satisfies $e_j(\mathcal{P}, h_\theta) \le v_j, 1 \le j \le L$. Then we characterize the properties (e.g., fair base rate) of the test distribution $\mathcal{Q}$: $g_j(\mathcal{Q}) \le u_j, 1 \le j \le M$. As a result, we can upper bound the loss of $h_\theta(\cdot)$ on $\mathcal{Q}$ as the following optimization:

$$\max_{\mathcal{Q}, \theta} \mathbb{E}_{(X,Y) \sim \mathcal{Q}}[\ell(h_\theta(X), Y)] \quad \text{s.t.} \quad H(\mathcal{P}, \mathcal{Q}) \le \rho, \quad e_j(\mathcal{P}, h_\theta) \le v_j \, \forall j \in [L], \quad g_j(\mathcal{Q}) \le u_j \, \forall j \in [M].$$
(2)

Now we decompose the space $\mathcal{Z} := \mathcal{X} \times \mathcal{Y}$ to $N$ partitions: $\mathcal{Z} := \biguplus \mathcal{Z}_i$, where $\mathcal{Z}$ is the support of both $\mathcal{P}$ and $\mathcal{Q}$. Then, we denote $\mathcal{P}$ conditioned on $\mathcal{Z}_i$ by $\mathcal{P}_i$ and similarly $\mathcal{Q}$ conditioned on $\mathcal{Z}_i$ by $\mathcal{Q}_i$. As a result, we can write $\mathcal{P} = \sum_{i \in [N]} p_i \mathcal{P}_i$ and $\mathcal{Q} = \sum_{i \in [N]} q_i \mathcal{Q}_i$. Since $\mathcal{P}$ is known, $p_i$'s are known. In contrast, both $\mathcal{Q}_i$ and $q_i$'s are optimizable. Our key observation is that

$$H(\mathcal{P}, \mathcal{Q}) \le \rho \iff 1 - \rho^2 - \sum_{i=1}^{N} \sqrt{p_i q_i}(1 - H(\mathcal{P}_i, \mathcal{Q}_i)^2) \le 0$$
(3)

---

[2]$\delta(\mathcal{P}, \mathcal{Q}) = \sup_{A \in \mathcal{F}} |\mathcal{P}(A) - \mathcal{Q}(A)|$ where $\mathcal{F}$ is a $\sigma$-algebra of subsets of the sample space $\Omega$.

which leads to the following theorem.

**Theorem 1.** *The following constrained optimization upper bounds Equation (2):*

$$\max_{\mathcal{Q}_i, q_i, \rho_i, \theta} \quad \sum_{i=1}^{N} q_i \mathbb{E}_{(X,Y)\sim\mathcal{Q}_i}[\ell(h_\theta(X), Y)] \tag{4a}$$

$$\text{s.t.} \quad 1 - \rho^2 - \sum_{i=1}^{N} \sqrt{p_i q_i}(1 - \rho_i^2) \leq 0, \tag{4b}$$

$$H(\mathcal{P}_i, \mathcal{Q}_i) \leq \rho_i \quad \forall i \in [N], \quad \sum_{i=1}^{N} q_i = 1, \quad q_i \geq 0 \quad \forall i \in [N], \quad \rho_i \geq 0 \quad \forall i \in [N], \tag{4c}$$

$$e'_j(\{\mathcal{P}_i\}_{i\in[N]}, \{p_i\}_{i\in[N]}, h_\theta) \leq v'_j \,\forall j \in [L], \quad g'_j(\{\mathcal{Q}_i\}_{i\in[N]}, \{q_i\}_{i\in[N]}) \leq u'_j \,\forall j \in [M], \tag{4d}$$

*if $e_j(\mathcal{P}, h_\theta) \leq v_j$ implies $e'_j(\{\mathcal{P}_i\}_{i\in[N]}, \{p_i\}_{i\in[N]}, h_\theta) \leq v'_j$ for any $j \in [L]$, and $g_j(\mathcal{Q}) \leq u_j$ implies $g'_j(\{\mathcal{Q}_i\}_{i\in[N]}, \{q_i\}_{i\in[N]}) \leq u'_j$ for any $j \in [M]$.*

In Problem 2, the challenge is to deal with the fair base rate constraint. Our core technique in Thm. 1 is subpopulation decomposition. At a high level, thanks to the disjoint support among different subpopulations, we get Equation (3). This equation gives us an equivalence relationship between distribution-level (namely, $\mathcal{P}$ and $\mathcal{Q}$) distance constraint and subpopulation-level (namely, $\mathcal{P}_i$'s and $\mathcal{Q}_i$'s) distance constraint. As a result, we can rewrite the original problem (2) using sub-population as decision variables as in Equation (4b) and then imposing the unity constraint (Equation (4c)) to get Thm. 1. We provide a detailed proof in Appendix C.2. Although the optimization problem (Equation (4)) may look more complicated then the original Equation (2), this optimization simplifies the challenging fair base rate constraint, allows us to upper bound each subpopulation loss $\mathbb{E}_{(X,Y)\sim\mathcal{Q}_i}[\ell(h_\theta(X), Y)]$ individually, and hence makes the whole optimization tractable.

### 3.2 Certified Fairness with Sensitive Shifting

For the sensitive shifting case, we instantiate Thm. 1 and obtain the following fairness certificate.

**Theorem 2.** *Given a distance bound $\rho > 0$, the following constrained optimization, which is **convex**, when feasible, provides a **tight** fairness certificate for Problem 2:*

$$\max_{k_s, r_y} \quad \sum_{s=1}^{S}\sum_{y=1}^{C} k_s r_y E_{s,y}, \quad \text{s.t.} \quad \sum_{s=1}^{S} k_s = 1, \quad \sum_{y=1}^{C} r_y = 1, \quad k_s \geq 0 \quad \forall s \in [S], \quad r_y \geq 0 \quad \forall y \in [C],$$

$$1 - \rho^2 - \sum_{s=1}^{S}\sum_{y=1}^{C} \sqrt{p_{s,y} k_s r_y} \leq 0,$$

*where $E_{s,y} := \mathbb{E}_{(X,Y)\sim\mathcal{P}_{s,y}}[\ell(h_\theta(X), Y)]$ and $p_{s,y} := \Pr_{(X,Y)\in\mathcal{P}}[X_s = s, Y = y]$ are constants.*

*Proof sketch.* We decompose distribution $\mathcal{P}$ and $\mathcal{Q}$ to $\mathcal{P}_{s,y}$'s and $\mathcal{Q}_{s,y}$'s according to their sensitive attribute and label values. In sensitive shifting, $\Pr(X_o|Y, X_s)$ is fixed, i.e., $\mathcal{P}_{s,y} = \mathcal{Q}_{s,y}$, which means $\mathbb{E}_{(X,Y)\sim\mathcal{Q}_{s,y}}[\ell(h_\theta(X), Y)] = E_{s,y}$ and $\rho_{s,y} = H(\mathcal{P}_{s,y}, \mathcal{Q}_{s,y}) = 0$. We plug these properties into Thm. 1. Then, denoting $q_{s,y}$ to $\Pr_{(X,Y)\sim\mathcal{Q}}[X_s = s, Y = y]$, we can represent the fairness constraint in Def. 2 as $q_{s_0,y_0} = \left(\sum_{s=1}^{S} q_{s,y_0}\right)\left(\sum_{y=1}^{C} q_{s_0,y}\right)$ for any $s_0 \in [S]$ and $y_0 \in [C]$. Next, we parameterize $q_{s,y}$ with $k_s r_y$. Such parameterization simplifies the fairness constraint and allow us to prove the convexity of the resulting optimization. Since all the constraints are encoded equivalently, the problem formulation provides a tight certification. Detailed proof in Appendix C.3. □

As Thm. 2 suggests, we can exploit the expectation information $E_{s,y} = \mathbb{E}_{(X,Y)\sim\mathcal{P}_{s,y}}[\ell(h_\theta(X), Y)]$ and density information $p_{s,y} = \Pr_{(X,Y)\sim\mathcal{P}}[X_s = s, Y = y]$ of each $\mathcal{P}$'s subpopulation to provide a tight fairness certificate in sensitive shifting. The convex optimization problem with $(S+C)$ variables can be efficiently solved by off-the-shelf packages.

### 3.3 Certified Fairness with General Shifting

For the general shifting case, we leverage Thm. 1 and the parameterization trick $q_{s,y} := k_s r_y$ used in Thm. 2 to reduce Problem 1 to the following constrained optimization.

**Lemma 3.1.** *Given a distance bound $\rho > 0$, the following constrained optimization, when feasible, provides a **tight** fairness certificate for Problem 1:*

$$\max_{k_s, r_y, \mathcal{Q}, \rho_{s,y}} \quad \sum_{s=1}^{S} \sum_{y=1}^{C} k_s r_y \mathbb{E}_{(X,Y) \sim \mathcal{Q}_{s,y}} [\ell(h_\theta(X), Y)] \tag{6a}$$

$$\text{s.t.} \quad \sum_{s=1}^{S} k_s = 1, \quad \sum_{y=1}^{C} r_y = 1, \quad k_s \geq 0 \quad \forall s \in [S], \quad r_y \geq 0 \quad \forall y \in [C], \tag{6b}$$

$$\sum_{s=1}^{S} \sum_{y=1}^{C} \sqrt{p_{s,y} k_s r_y} (1 - \rho_{s,y}^2) \geq 1 - \rho^2 \tag{6c}$$

$$H(\mathcal{P}_{s,y}, \mathcal{Q}_{s,y}) \leq \rho_{s,y} \quad \forall s \in [S], y \in [C], \tag{6d}$$

*where $p_{s,y} := \Pr_{(X,Y) \in \mathcal{P}}[X_s = s, Y = y]$ is a fixed constant. The $\mathcal{P}_{s,y}$ and $\mathcal{Q}_{s,y}$ are the subpopulations of $\mathcal{P}$ and $\mathcal{Q}$ on the support $\{(X,Y) : X \in \mathcal{X}, X_s = s, Y = y\}$ respectively.*

*Proof sketch.* We show that Equation (6b) ensures a parameterization of $q_{s,y} = \Pr_{(X,Y) \in \mathcal{Q}}[X_s = s, Y = y]$ that satisfies fairness constraints on $\mathcal{Q}$. Then, leveraging Thm. 1 we prove that the constrained optimization provides a fairness certificate. Since all the constraints are either kept or equivalently encoded, this resulting certification is *tight*. Detailed proof in Appendix C.4. □

Now the main obstacle is to solve the non-convex optimization in Problem 6. Here, as the first step, we upper bound the loss of $h_\theta(\cdot)$ within each shifted subpopulation $\mathcal{Q}_{s,y}$, i.e., upper bound $\mathbb{E}_{(X,Y) \sim \mathcal{Q}_{s,y}}[\ell(h_\theta(X), Y)]$ in Equation (6a), by Thm. 4 in Appendix B.2 [47]. Then, we apply variable transformations to make some decision variables convex. For the remaining decision variables, we observe that they are non-convex but bounded. Hence, we propose the technique of grid-based sub-problem construction. Concretely, we divide the feasible region regarding non-convex variables into small grids and consider the optimization problem in each region individually. For each sub-problem, we relax the objective by pushing the values of non-convex variables to the boundary of the current grid and then solve the convex optimization sub-problems. Concretely, the following theorem states our computable certificate for Problem 1, with detailed proof in Appendix C.5.

**Theorem 3.** *If for any $s \in [S]$ and $y \in [Y]$, $H(\mathcal{P}_{s,y}, \mathcal{Q}_{s,y}) \leq \bar{\gamma}_{s,y}$ and $0 \leq \sup_{(X,Y) \in \mathcal{X} \times \mathcal{Y}} \ell(h_\theta(X), Y) \leq M$, given a distance bound $\rho > 0$, for any region granularity $T \in \mathbb{N}_+$, the following expression provides a fairness certificate for Problem 1:*

$$\bar{\ell} = \max_{\{i_s \in [T]: s \in [S]\}, \{j_y \in [T]: y \in [C]\}} \mathbf{C}\left(\left\{\left[\frac{i_s - 1}{T}, \frac{i_s}{T}\right]\right\}_{s=1}^{S}, \left\{\left[\frac{j_y - 1}{T}, \frac{j_y}{T}\right]\right\}_{y=1}^{C}\right), \text{ where} \tag{7}$$

$$\mathbf{C}\left(\{[\underline{k_s}, \overline{k_s}]\}_{s=1}^{S}, \{[\underline{r_y}, \overline{r_y}]\}_{y=1}^{C}\right) = \max_{x_{s,y}} \sum_{s=1}^{S} \sum_{y=1}^{C} \left(\overline{k_s} \overline{r_y} (E_{s,y} + C_{s,y})_+ + \underline{k_s} \underline{r_y} (E_{s,y} + C_{s,y})_- \right.$$

$$\left. + 2\overline{k_s} \overline{r_y} \sqrt{x_{s,y}(1 - x_{s,y})} \sqrt{V_{s,y}} - \underline{k_s} \underline{r_y} x_{s,y} (C_{s,y})_+ - \overline{k_s} \overline{r_y} x_{s,y} (C_{s,y})_- \right) \tag{8a}$$

$$\text{s.t.} \quad \sum_{s=1}^{S} \underline{k_s} \leq 1, \quad \sum_{s=1}^{S} \overline{k_s} \geq 1, \quad \sum_{y=1}^{C} \underline{r_y} \leq 1, \quad \sum_{y=1}^{C} \overline{r_y} \geq 1, \tag{8b}$$

$$\sum_{s=1}^{S} \sum_{y=1}^{C} \sqrt{p_{s,y} \overline{k_s r_y} x_{s,y}} \geq 1 - \rho^2, \quad (1 - \bar{\gamma}_{s,y}^2)^2 \leq x_{s,y} \leq 1 \quad \forall s \in [S], y \in [C], \tag{8c}$$

*where $(\cdot)_+ = \max\{\cdot, 0\}$, $(\cdot)_- = \min\{\cdot, 0\}$; $E_{s,y} = \mathbb{E}_{(X,Y) \sim \mathcal{P}_{s,y}}[\ell(h_\theta(X), Y)]$, $V_{s,y} = \mathbb{V}_{(X,Y) \sim \mathcal{P}_{s,y}}[\ell(h_\theta(X), Y)]$, $p_{s,y} = \Pr_{(X,Y) \sim \mathcal{P}}[X_s = s, Y = y]$, $C_{s,y} = M - E_{s,y} - \frac{V_{s,y}}{M - E_{s,y}}$, and $\bar{\gamma}_{s,y}^2 = 1 - (1 + (M - E_{s,y})^2 / V_{s,y})^{-\frac{1}{2}}$. Equation (7) only takes $\mathbf{C}$'s value when it is feasible, and each $\mathbf{C}$ queried by Equation (7) is a **convex optimization**.*

**Implications.** Thm. 3 provides a fairness certificate for Problem 1 under two assumptions: (1) The loss function is bounded (by $M$). This assumption holds for several typical losses such as 0-1 loss and JSD loss. (2) The distribution shift between training and test distribution within each subpopulation is bounded by $\bar{\gamma}_{s,y}$, where $\bar{\gamma}_{s,y}$ is determined by the model's statistics on $\mathcal{P}$. In practice, this additional distance bound assumption generally holds, since $\bar{\gamma}_{s,y} \gg \rho$ for common choices of $\rho$.

In Thm. 3, we exploit three types of statistics of $h_\theta(\cdot)$ on $\mathcal{P}$ to compute the fairness certificates: the expectation $E_{s,y} = \mathbb{E}_{(X,Y)\sim\mathcal{P}_{s,y}}[\ell(h_\theta(X),Y)]$, the variance $V_{s,y} = \mathbb{V}_{(X,Y)\sim\mathcal{P}_{s,y}}[\ell(h_\theta(X),Y)]$, and the density $p_{s,y} = \Pr_{(X,Y)\sim\mathcal{P}}[X_s = s, Y = y]$, all of which are at the subpopulation level and a high-confidence estimation of them based on finite samples are tractable (Section 3.4).

Using Thm. 3, after determining the region granularity $T$, we can provide a fairness certificate for Problem 1 by solving $T^{SC}$ convex optimization problems, each of which has $SC$ decision variables. Note that the computation cost is independent of $h_\theta$, and therefore we can numerically compute the certificate for large DNN models used in practice. Specifically, when $S = 2$ (binary sensitive attribute) or $C = 2$ (binary classification) which is common in the fairness evaluation setting, we can construct the region for only one dimension $k_1$ or $r_1$, and use $1 - k_1$ or $1 - r_1$ for the other dimension. Thus, for the typical setting $S = 2, C = 2$, we only need to solve $T^2$ convex optimization problems.

Note that for Problem 2, our certificate in Thm. 2 is tight, whereas for Problem 1, our certificate in Thm. 3 is not. This is because in Problem 1, extra distribution shift exists within each subpopulation, i.e., $\Pr(X_o|Y, X_s)$ changes from $\mathcal{P}$ to $\mathcal{Q}$, and to bound such shift, we need to leverage Thm. 2.2 in [47] which has no tightness guarantee. Future work providing tighter bounds than [47] can be seamlessly incorporated into our framework to tighten our fairness certificate for Problem 1.

### 3.4 Dealing with Finite Sampling Error

In Section 3.2 and Section 3.3, we present Thm. 2 and Thm. 3 that provide computable fairness certificates for sensitive shifting and general shifting scenarios respectively. In these theorems, we need to know the quantities related to the training distribution and trained $\mathcal{P}$ and model $h_\theta(\cdot)$:

$$E_{s,y} = \mathbb{E}_{(X,Y)\sim\mathcal{P}_{s,y}}[\ell(h_\theta(X),Y)], V_{s,y} = \mathbb{V}_{(X,Y)\sim\mathcal{P}_{s,y}}[\ell(h_\theta(X),Y)], p_{s,y} = \Pr_{(X,Y)\sim\mathcal{P}}[X_s = s, Y = y]. \quad (9)$$

Section 3.3 further requires $C_{s,y}$ and $\bar{\gamma}_{s,y}$ which are functions of $E_{s,y}$ and $V_{s,y}$. However, a practical challenge is that common training distributions do not have an analytical expression that allows us to precisely compute these quantities. Indeed, we only have access to a finite number of individually drawn samples, i.e., the training dataset, from $\mathcal{P}$. Thus, we will provide high-confidence bounds for $E_{s,y}$, $V_{s,y}$, and $p_{s,y}$ in Lemma D.1 (stated in Appendix D.1).

For Thm. 2, we can replace $E_{s,y}$ in the objective by the upper bounds of $E_{s,y}$ and replace the concrete quantities of $p_{s,y}$ by interval constraints and the unit constraint $\sum_s \sum_y p_{s,y} = 1$, which again yields a convex optimization that can be effectively solved. For Thm. 3, we compute the confidence intervals of $C_{s,y}$ and $\rho_{s,y}$, then plug in either the lower bounds or the upper bounds to the objective (8a) based on the coefficient, and finally replace the concrete quantities of $p_{s,y}$ by interval constraints and the unit constraint $\sum_s \sum_y p_{s,y} = 1$. The resulting optimization is proved to be convex and provides an upper bound for any possible values of $E_{s,y}$, $V_{s,y}$, and $p_{s,y}$ within the confidence intervals. We defer the statement of Thm. 2 and Thm. 3 considering finite sampling error to Appendix D.2. To this point, we have presented our framework for computing high-confidence fairness certificates given access to model $h_\theta(\cdot)$ and a finite number of samples drawn from $\mathcal{P}$.

## 4 Experiments

In this section, we evaluate the certified fairness under both *sensitive shifting* and *general shifting* scenarios on six real-world datasets. We observe that under the sensitive shifting, our certified fairness bound is *tight* (Section 4.1); while the bound is less tight under general shifting (Section 4.2) which depends on the tightness of generalization bounds within each subpopulation (details in Section 3.3). In addition, we show that our certification framework can flexibly integrate more constraints on $\mathcal{Q}$, leading to a tighter fairness certification (Section 4.3). Finally, we compare our certified fairness bound with existing distributional robustness bound [43] (section 4.4), since both consider a shifted distribution while our bound is optimized with an additional fairness constraint which is challenging to be directly integrated to the existing distributional robustness optimization. We show that with the fairness constraint and our optimization approach, our bound is much tighter.

**Dataset & Model.** We validate our certified fairness on *six* real-world datasets: Adult [3], Compas [2], Health [19], Lawschool [48], Crime [3], and German [3]. Details on the datasets and data processing steps are provided in Appendix E.1. Following the standard setup of fairness evaluation in the literature [39, 38, 31, 42], we consider the scenario that the sensitive attributes and labels take binary values. The ReLU network composed of 2 hidden layers of size 20 is used for all datasets.

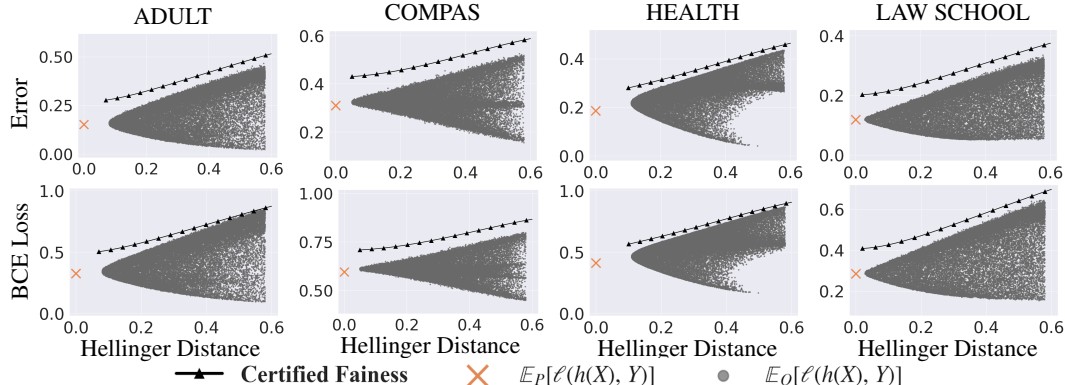

Figure 1: Certified fairness with sensitive shifting. Grey points are results on generated distributions ($\mathcal{Q}$) and the black line is our fairness certificate based on Thm. 2. We observe that our fairness certificate is usually tight.

**Fairness Certification.** We perform vanilla model training and then leverage our fairness certification framework to calculate the fairness certificate. Concretely, we input the trained model information on $\mathcal{P}$ and the framework would output the fairness certification for both sensitive shifting and general shifting scenarios following Thm. 2 and Thm. 3, respectively.

Code, model, and all experimental data are publicly available at `https://github.com/AI-secure/Certified-Fairness`.

### 4.1 Certified Fairness with Sensitive Shifting

**Generating Fair Distributions.** To evaluate how well our certificates capture the fairness risk in practice, we compare our certification bound with the empirical loss evaluated on randomly generated $30,000$ fairness constrained distributions $\mathcal{Q}$ shifted from $\mathcal{P}$. The detailed steps for generating fairness constrained distributions $\mathcal{Q}$ are provided in Appendix E.2. Under sensitive shifting, since each subpopulation divided by the sensitive attribute and label does not change (Section 2.1), we tune only the portion of each subpopulation $q_{s,y}$ satisfying the base rate fairness constraint, and then sample from each subpopulation of $\mathcal{P}$ individually according to the proportion $q_{s,y}$. In this way, our protocols can generate distributions with different combinations of subpopulation portions. If the classifier is biased toward one subpopulation (i.e., it achieves high accuracy in the group but low accuracy in others), the worst-case accuracy on generated distribution is low since the portion of the biased subpopulation in the generated distribution can be low; in contrast, a fair classifier which performs uniformly well for each group can achieve high worst-case accuracy (high certified fairness). Therefore, we believe that our protocols can demonstrate real-world training distribution bias as well as reflect the model's unfairness and certification tightness in real-world scenarios.

**Results.** We report the classification error (Error) and BCE loss as the evaluation metric. Figure 1 illustrates the certified fairness on Adult, Compas, Health, and Lawschool under sensitive shifting. More results on two relatively small datasets (Crime, German) are shown in Appendix E.5. From the results, we see that our certified fairness is tight in practice.

### 4.2 Certified Fairness with General Shifting

In the general shifting scenario, we similarly randomly generate $30,000$ fair distributions $\mathcal{Q}$ shifted from $\mathcal{P}$. Different from sensitive shifting, the distribution conditioned on sensitive attribute $X_s$ and label $Y$ can also change in this scenario. Therefore, we construct another distribution $\mathcal{Q}'$ disjoint with $\mathcal{P}$ on non-sensitive attributes and mix $\mathcal{P}$ and $\mathcal{Q}'$ in each subpopulation individually guided by mixing parameters satisfying fair base rate constraint. Detailed generation steps are given in Appendix E.2. Since the fairness certification for general shifting requires bounded loss, we select classification error (Error) and Jensen-Shannon loss (JSD Loss) as the evaluation metric. Figure 2 illustrates the certified fairness with classification error metric under general shifting. Results of JSD loss and more results on two relatively small datasets (Crime, German) are in Appendix E.5.

### 4.3 Certified Fairness with Additional Non-Skewness Constraints

In Section 2.1, we discussed that to represent different real-world scenarios we can add more constraints such as Equation (1) to prevent the skewness of $\mathcal{Q}$, which can be flexibly incorporated into our certificate framework. Concretely, for sensitive shifting, we only need to add one more

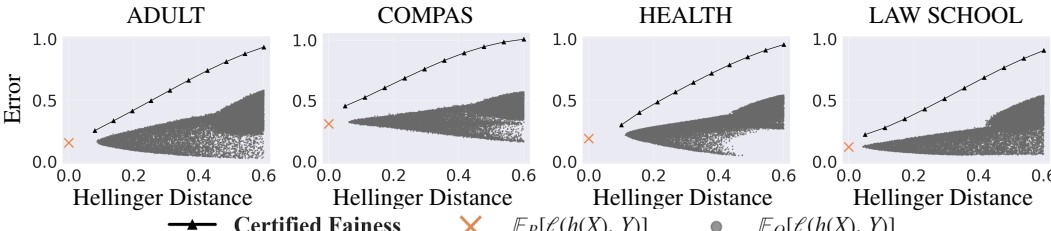

Figure 2: Certified fairness with general shifting. Grey points are results on generated distributions ($\mathcal{Q}$) and the black line is our fairness certificate based on Thm. 3. We observe that our fairness certificate is non-trivial.

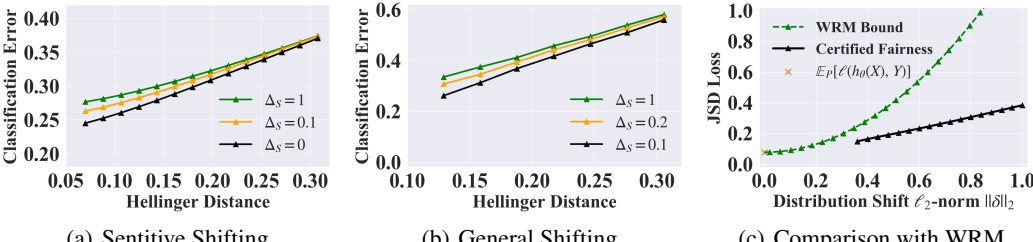

(a) Sentitive Shifting        (b) General Shifting        (c) Comparison with WRM

Figure 3: Certified fairness with additional non-skewness constraints on Adult dataset is shown in (a) (b). $\Delta_s$ controls the skewness of $\mathcal{Q}$ ($|\Pr_{(X,Y)\sim\mathcal{Q}}[X_s = 0] - \Pr_{(X,Y)\sim\mathcal{Q}}[X_s = 1]| \leq \Delta_s$). More analysis in Section 4.3. In (c), we compare our certified fairness bound with the distributional robustness bound [43]. More analysis in Section 4.4.

box constraint[3] $0.5 - \Delta_s/2 \leq k_s \leq 0.5 + \Delta_s/2$ where $\Delta_s$ is a parameter controlling the skewness of $\mathcal{Q}$, which still guarantees convexity. For general shifting, we only need to modify the region partition step[3], where we split $[0.5 - \Delta_s/2, 0.5 + \Delta_s/2]$ instead of $[0, 1]$. The certification results with additional constraints are in Figures 3(a) and 3(b), which suggests that if the added constraints are strict (i.e., smaller $\Delta_s$), the bound is tighter. More constraints w.r.t. labels can also be handled by our framework and the corresponding results as well as results on more datasets are in Appendix E.6.

## 4.4 Comparison with Distributional Robustness Bound

To the best of our knowledge, there is no existing work providing *certified fairness* on the end-to-end model performance. Thus, we try to compare our bound with the distributional robustness bound since both consider certain distribution shifts. However, it is challenging to directly integrate the fairness constraints into existing bounds. Therefore, we compare with the state-of-the-art distributional robustness certification WRM [43], which solves the similar optimization problem as ours except for the fairness constraint. For fair comparison, we construct a synthetic dataset following [43], on which there is a one-to-one correspondence between the Hellinger and Wasserstein distance used by WRM. We randomly select one dimension as the sensitive attribute. Since WRM has additional assumptions on smoothness of models and losses, we use JSD loss and a small ELU network with 2 hidden layers of size 4 and 2 following their setting. More implementation details are in Appendix E.4. Results in Figure 3(c) suggest that 1) our certified fairness bound is much tighter than WRM given the additional fairness distribution constraint and our optimization framework; 2) with additional fairness constraint, our certificate problem could be infeasible under very small distribution distances since the fairness constrained distribution $\mathcal{Q}$ does not exist near the skewed original distribution $\mathcal{P}$; 3) with the fairness constraint, we provide non-trivial fairness certification bound even when the distribution shift is large.

## 5 Conclusion

In this paper, we provide the first *fairness certification* on end-to-end model performance, based on a fairness constrained distribution which has bounded distribution distance from the training distribution. We show that our fairness certification has strong connections with existing fairness notions such as group parity, and we provide an effective framework to calculate the certification under different scenarios. We provide both theoretical and empirical analysis of our fairness certification.

**Acknowledgements.** MK, LL, and BL are partially supported by the NSF grant No.1910100, NSF CNS No.2046726, C3 AI, and the Alfred P. Sloan Foundation. YL is partially supported by the NSF grants IIS-2143895 and IIS-2040800.

---

[3]Note that such modification is only viable when sentive attributes take binary values, which is the typical scenario in the literature of fairness evaluation [39, 38, 31, 42].

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
