# Appendices

## Contents

# A   Broader Impact

This paper aims to calculate a *fairness certificate* under some distributional fairness constraints on the performance of an end-to-end ML model. We believe that the rigorous fairness certificates provided by our framework will significantly benefit and advance social fairness in the era of deep learning. Especially, such fairness certificate can be directly used to measure the fairness of an ML model regardless the target domain, which means that it will measure the unique property of the model itself with theoretical guarantees, and thus help people understand the risks of existing ML models. As a result, the ML community may develop ML training algorithms that explicitly reduce the fairness risks by regularizing on this fairness certificate.

A possible negative societal impact may stem from the misunderstanding or inaccurate interpretation of our fairness certificate. As a first step towards distributional fairness certification, we define the fairness through the lens of worst-case performance loss on a fairness constrained distribution. This fairness definition may not explicitly imply an absolute fairness guarantee under some other criterion. For example, it does not imply that for any possible individual input, the ML model will give fair prediction. We tried our best in Section 2 to define the certification goal, and the practitioners may need to understand this goal well to avoid misinterpretation or misuse of our fairness certification.

# B   Omitted Background

We illustrate omitted background in this appendix.

## B.1   Hellinger Distance

As illustrated in the beginning of Section 3, our framework uses Hellinger distance to bound the distributional distance. A formal definition of Hellinger distance is as below.

**Definition 3** (Hellinger Distance). *Let $\mathcal{P}$ and $\mathcal{Q}$ be distributions on $\mathcal{Z} := \mathcal{X} \times \mathcal{Y}$ that are absolutely continuous with respect to a reference measure $\mu$ with $\mathcal{P}, \mathcal{Q} \ll \mu$. The Hellinger distance between $\mathcal{P}$ and $\mathcal{Q}$ is defined as*

$$H(\mathcal{P}, \mathcal{Q}) := \sqrt{\frac{1}{2} \int_{\mathcal{Z}} \left( \sqrt{p(z)} - \sqrt{q(z)} \right)^2 \mathrm{d}\mu(z)} \tag{10}$$

*where $p = \frac{d\mathcal{P}}{d\mu}$ and $q = \frac{d\mathcal{Q}}{d\mu}$ are the Radon-Nikodym derivatives of $\mathcal{P}$ and $\mathcal{Q}$ with respect to $\mu$, respectively. The Hellinger distance is independent of the choice of the reference measure $\mu$.*

Representative properties for the Hellinger distance are discussed in Section 3.

## B.2   Thm. 2.2 in [47]

As mentioned in Section 3.3, we leverage Thm. 2.2 from [47] to upper bound the expected loss of $h_\theta(\cdot)$ in each shifted subpopulation $\mathcal{Q}_{s,y}$. Here we restate Thm. 2.2 for completeness.

**Theorem 4** (Thm. 2.2, [47]). *Let $\mathcal{P}'$ and $\mathcal{Q}'$ denote two distributions supported on $\mathcal{X} \times \mathcal{Y}$, suppose that $0 \le \ell(h_\theta(X), Y) \le M$, then*

$$\max_{\mathcal{Q}', \theta} \mathbb{E}_{(X,Y) \sim \mathcal{Q}'}[\ell(h_\theta(X), Y)] \quad \text{s.t.} \quad H(\mathcal{P}', \mathcal{Q}') \le \rho$$

$$\le \mathbb{E}_{(X,Y) \sim \mathcal{P}'}[\ell(h_\theta(X), Y)] + 2C_\rho \sqrt{\mathbb{V}_{(X,Y) \sim \mathcal{P}'}[\ell(h_\theta(X), Y)]} +$$

$$\rho^2(2 - \rho^2) \left( M - \mathbb{E}_{(X,Y) \sim \mathcal{P}'}[\ell(h_\theta(X), Y)] - \frac{\mathbb{V}_{(X,Y) \sim \mathcal{P}'}[\ell(h_\theta(X), Y)]}{M - \mathbb{E}_{(X,Y) \sim \mathcal{P}'}[\ell(h_\theta(X), Y)]} \right), \tag{11}$$

*where $C_\rho = \sqrt{\rho^2(1 - \rho^2)^2(2 - \rho^2)}$, for any given distance bound $\rho > 0$ that satisfies*

$$\rho^2 \le 1 - \left( 1 + \frac{(M - \mathbb{E}_{(X,Y) \sim \mathcal{P}'}[\ell(h_\theta(X), Y)])^2}{\mathbb{V}_{(X,Y) \sim \mathcal{P}'}[\ell(h_\theta(X), Y)]} \right)^{-1/2}. \tag{12}$$

This theorem provides a closed-form expression that upper bounds the mean loss of $h_\theta(\cdot)$ on shifted distribution (namely $\mathbb{E}_{\mathcal{Q}'}[\ell(h_\theta(X), Y)]$), given bounded Hellinger distance $H(\mathcal{P}, \mathcal{Q})$ and the mean $E$ and variance $V$ of loss on $\mathcal{P}$ under two mild conditions: (1) the function is positive and bounded (denote the upper bound by $M$); and (2) the distance $H(\mathcal{P}, \mathcal{Q})$ is not too large (specifically, $H(\mathcal{P}, \mathcal{Q})^2 \leq \bar{\gamma}^2 := 1 - (1 + (M - E)^2/V)^{-\frac{1}{2}})$. Since Thm. 4 holds for arbitrary models and loss functions $\ell(h_\theta(\cdot), \cdot)$ as long as the function value is bounded by $[0, M]$, using Thm. 4 allows us to provide a generic and succinct fairness certificate in Thm. 3 for general shifting case that holds for generic models including DNNs without engaging complex model architectures. Indeed, we only need to query the mean and variance under $\mathcal{P}$ for the given model to compute the certificate in Thm. 4, and this benefit is also inherited by our certification framework expressed by Thm. 3. Note that there is no tightness guarantee for this bound yet, which is also inherited by our Thm. 3.

## C    Proofs of Main Results

This appendix entails the complete proofs for Proposition 1, Thm. 1, Thm. 2, Lemma 3.1, and Thm. 3 in the main text. For complex proofs such as that for Thm. 3, we also provide high-level illustration before going into the formal proof.

### C.1    Proof of Proposition 1

*Proof of Proposition 1.* Since each term $\Pr_{(X,Y)\sim\mathcal{Q}}[h_\theta(X) \neq Y | Y = y, X_s = i]$ is within $[0, \epsilon]$, we consider two cases: $y \neq 1$ and $y = 1$. If $y \neq 1$, $\Pr_{(X,Y)\sim\mathcal{Q}}[h_\theta(X) = 1 | Y = y, X_s = i] \leq \Pr_{(X,Y)\sim\mathcal{Q}}[h_\theta(X) \neq Y | Y = y, X_s = i] \leq \epsilon$ and so will be their differences for $X_s = i$ and $X_s = j$. If $y = 1$, $\Pr_{(X,Y)\sim\mathcal{Q}}[h_\theta(X) = 1 | Y = y, X_s = i] = 1 - \Pr_{(X,Y)\sim\mathcal{Q}}[h_\theta(X) \neq Y | Y = y, X_s = i] \in [1 - \epsilon, 1]$, and also the differences for $X_s = i$ and $X_s = j$ are always within $\epsilon$. This proves $\epsilon$-EO.

Now consider DP. We notice that for any $a$,

$$\Pr_{(X,Y)\sim\mathcal{Q}}[h_\theta(X) = 1 | X_s = a] = \sum_{y=1}^{C} \Pr_{(X,Y)\sim\mathcal{Q}}[h_\theta(X) = 1 | Y = y, X_s = a] \cdot \Pr_{(X,Y)\sim\mathcal{Q}}[Y = y | X_s = a].$$

(13)

Thus,

$$\left| \Pr_{(X,Y)\sim\mathcal{Q}}[h_\theta(X) = 1 | X_s = i] - \Pr_{(X,Y)\sim\mathcal{Q}}[h_\theta(X) = 1 | X_s = j] \right|$$

$$\overset{(*)}{\leq} \sum_{y=1}^{C} \left| \Pr_{(X,Y)\sim\mathcal{Q}}[h_\theta(X) = 1 | Y = y, X_s = i] - \Pr_{(X,Y)\sim\mathcal{Q}}[h_\theta(X) = 1 | Y = y, X_s = j] \right|$$

$$\cdot \Pr_{(X,Y)\sim\mathcal{Q}}[Y = y | X_s = i]$$

$$\leq \sum_{y=1}^{C} \epsilon \Pr_{(X,Y)\sim\mathcal{Q}}[Y = y | X_s = i] = \epsilon$$

which proves $\epsilon$-DP, where $(*)$ leverages the fair base rate property of $\mathcal{Q}$ which gives $\Pr_{(X,Y)\sim\mathcal{Q}}[Y = y | X_s = i] = \Pr_{(X,Y)\sim\mathcal{Q}}[Y = y | X_s = j]$. $\square$

### C.2    Proof of Thm. 1

*Proof of Thm. 1.* We first prove the key eq. (3).

$$H(\mathcal{P}, \mathcal{Q}) \leq \rho \iff H^2(\mathcal{P}, \mathcal{Q}) \leq \rho^2$$

$$\iff \frac{1}{2} \int_{\mathcal{Z}} \left( \sqrt{p(z)} - \sqrt{q(z)} \right)^2 \mathrm{d}\mu(z) \leq \rho^2$$

$$\iff \frac{1}{2} \left( \int_{\mathcal{Z}} p(z)\mathrm{d}\mu(z) + \int_{\mathcal{Z}} q(z)\mathrm{d}\mu(z) \right) - \int_{\mathcal{Z}} \sqrt{p(z)q(z)}\mathrm{d}\mu(z) \leq \rho^2$$

$$\iff \int_{\mathcal{Z}} \sqrt{p(z)q(z)}\mathrm{d}\mu(z) \geq 1 - \rho^2$$

$$\iff \sum_{i=1}^{N} \int_{\mathcal{Z}_i} \sqrt{p_i q_i} \cdot \sqrt{p_i(z)q_i(z)}\mathrm{d}\mu(z) \geq 1 - \rho^2$$

$$\iff \sum_{i=1}^{N} \sqrt{p_i q_i}\left(1 - H^2(\mathcal{P}_i, \mathcal{Q}_i)\right) \geq 1 - \rho^2 \tag{14}$$

where $p_i(\cdot)$ and $q_i(\cdot)$ are density functions of subpopulation distributions $\mathcal{P}_i$ and $\mathcal{Q}_i$ respectively.

Then, we show that any feasible solution of Equation (2) satisfies the constraints in Equation (4). We let $\mathcal{Q}^\star$ and $\theta^\star$ denote a feasible solution of Equation (2), i.e.,

$$H(\mathcal{P}, \mathcal{Q}^\star) \leq \rho, \quad e_j(\mathcal{P}, h_{\theta^\star}) \leq v_j \ \forall j \in [L], \quad g_j(\mathcal{Q}^\star) \leq u_j \ \forall j \in [M]. \tag{15}$$

We let $\{q_i^\star\}_{i=1}^{N}$ denote the proportions of $\mathcal{Q}^\star$ within each support partition $\mathcal{Z}_i$, and $\{\mathcal{Q}_i^\star\}_{i=1}^{N}$ the $\mathcal{Q}^\star$ in each subpopulation. By Equation (14), we have $1 - \rho^2 - \sum_{i=1}^{N} \sqrt{p_i q_i^\star}(1 - \rho_i^2) \leq 0$ where $\rho_i = H^2(\mathcal{P}_i, \mathcal{Q}_i^\star)$. Note that by definition, $\sum_{i=1}^{N} q_i^\star = 1$ and $\forall i \in [N], q_i^\star \geq 0, \rho_i \geq 0$. Furthermore, by the implication relations stated in Thm. 1, for any $j \in [L]$, $e_j'(\{\mathcal{P}_i\}_{i=1}^{N}, \{p_i\}_{i=1}^{N}, h_{\theta^\star}) \leq v_j'$; and for any $j \in [M]$, $g_j'(\{\mathcal{Q}_i^\star\}_{i=1}^{N}, \{q_i^\star\}_{i=1}^{N}) \leq u_j'$. To this point, we have shown $\mathcal{Q}^\star$ and $\theta^\star$ satisfy all constraints in Equation (4), i.e., $\mathcal{Q}^\star$ and $\theta^\star$ is a feasible solution of Equation (4). Since Equation (4) expresses the optimal (maximum) solution, Equation (4) (in Thm. 1) $\geq$ Equation (2). $\square$

### C.3 Proof of Thm. 2

*Proof of Thm. 2.* The proof of Thm. 2 is composed of three parts: (1) the optimization problem provides a fairness certificate for Problem 2; (2) the certificate is tight; and (3) the optimization problem is convex.

**(1)** Suppose the maximum of Problem 2 is attained with the test distribution $\mathcal{Q}^\star$ in the sensitive shifting setting, then we decompose both $\mathcal{P}$ and $\mathcal{Q}^\star$ according to both the sensitive attribute and the label:

$$\mathcal{P} = \sum_{s=1}^{S}\sum_{y=1}^{C} p_{s,y}\mathcal{P}_{s,y}, \quad \mathcal{Q}^\star = \sum_{s=1}^{S}\sum_{y=1}^{C} q_{s,y}^\star\mathcal{Q}_{s,y}^\star. \tag{16}$$

Since $\mathcal{Q}^\star$ is a fair base rate distribution, for any $i, j \in [S]$, $b_{i,y}^{\mathcal{Q}^\star} = b_{j,y}^{\mathcal{Q}^\star}$ where $b_{s,y}^{\mathcal{Q}^\star} = \Pr_{(X,Y)\sim\mathcal{Q}^\star}[Y = y | X_s = s]$. As a result, $\Pr_{(X,Y)\sim\mathcal{Q}^\star}[Y = y | X_s = s] = \Pr_{(X,Y)\sim\mathcal{Q}^\star}[Y = y]$. Now we define

$$k_s^\star := \Pr_{(X,Y)\sim\mathcal{Q}^\star}[X_s = s], \quad r_y^\star := \Pr_{(X,Y)\sim\mathcal{Q}^\star}[Y = y], \tag{17}$$

and then

$$q_{s,y}^\star = \Pr_{(X,Y)\sim\mathcal{Q}^\star}[X_s = s, Y = y] = \Pr_{(X,Y)\sim\mathcal{Q}^\star}[X_s = s] \cdot \Pr_{(X,Y)\sim\mathcal{Q}^\star}[Y = y | X_s = s] = k_s^\star r_y^\star. \tag{18}$$

By the distance constraint in Problem 2 (namely $H(\mathcal{P}, \mathcal{Q}^\star) \leq \rho$) and Equation (14), we have

$$\sum_{s=1}^{S}\sum_{y=1}^{C} \sqrt{p_{s,y}q_{s,y}^\star}\left(1 - H^2(\mathcal{P}_{s,y}, \mathcal{Q}_{s,y}^\star)\right) \geq 1 - \rho^2. \tag{19}$$

Since there is only sensitive shifting, $H^2(\mathcal{P}_{s,y}, \mathcal{Q}_{s,y}^\star) = 0$, given Equation (18), we have

$$\sum_{s=1}^{S}\sum_{y=1}^{C} \sqrt{p_{s,y}k_s^\star r_y^\star} \geq 1 - \rho^2. \tag{20}$$

Now, we can observe that the $k_s^\star$ and $r_y^\star$ induced by $\mathcal{Q}^\star$ satisfy all constraints of Problem 2. For the objective,

Objective in Thm. 2

$$= \sum_{s=1}^{S} \sum_{y=1}^{C} k_s^\star r_s^\star \mathbb{E}_{(X,Y)\sim\mathcal{P}_{s,y}}[\ell(h_\theta(X), Y)]$$

$$= \sum_{s=1}^{S} \sum_{y=1}^{C} q_{s,y}^\star \mathbb{E}_{(X,Y)\sim\mathcal{Q}_{s,y}^\star}[\ell(h_\theta(X), Y)] \quad \text{(by Equation (18) and } H^2(\mathcal{P}_{s,y}, \mathcal{Q}_{s,y}^\star) = 0)$$

$$= \mathbb{E}_{(X,Y)\sim\mathcal{Q}^\star}[\ell(h_\theta(X), Y)]$$

$$= \text{Optimal value of Problem 2.}$$

Therefore, the *optimal* value of Thm. 2 will be larger or equal to the optimal value of Problem 2 which concludes the proof of the first part.

(2) Suppose the optimal value of Thm. 2 is attained with $k_s^\star$ and $r_y^\star$. We then construct $\mathcal{Q}^\star = \sum_{s=1}^{S} \sum_{y=1}^{C} k_s^\star r_y^\star \mathcal{P}_{s,y}$. We now inspect each constraint of Problem 2. The constraint $\text{dist}(\mathcal{P}, \mathcal{Q}^\star) \leq \rho$ is satisfied because $1 - \rho^2 - \sum_{s=1}^{S} \sum_{y=1}^{C} \sqrt{p_{s,y} k_s^\star r_y^\star} \leq 0$ is satisfied as a constraint of Thm. 2. Apparently, $\mathcal{P}_{s,y} = \mathcal{Q}_{s,y}^\star$. Then, $\mathcal{Q}^\star$ is a fair base rate distribution because

$$b_{s,y}^{\mathcal{Q}^\star} = \Pr_{(X,Y)\sim\mathcal{Q}^\star}[Y = y | X_s = s] = \frac{k_s^\star r_y^\star}{k_s^\star} = r_y^\star \tag{21}$$

is a constant across all $s \in [S]$. Thus, $\mathcal{Q}^\star$ satisfies all constraints of Problem 2 and

Optimal objective of Problem 2

$$\geq \mathbb{E}_{(X,Y)\sim\mathcal{Q}^\star}[\ell(h_\theta(X), Y)]$$

$$= \sum_{s=1}^{S} \sum_{y=1}^{C} k_s^\star r_y^\star \mathbb{E}_{(X,Y)\sim\mathcal{P}_{s,y}}[\ell(h_\theta(X), Y)] \tag{22}$$

$$= \sum_{s=1}^{S} \sum_{y=1}^{C} k_s^\star r_y^\star E_{s,y} = \text{Optimal objective of Thm. 2.}$$

Combining with the conclusion of the first part, we know optimal values of Thm. 2 and Problem 2 match, i.e., the certificate is tight.

(3) Inspecting the problem definition in Thm. 2, we find the objective and all constraints but the last one are linear. Therefore, to prove the convexity of the optimization problem, we only need to show that the last constraint

$$1 - \rho^2 - \sum_{s=1}^{S} \sum_{y=1}^{C} \sqrt{p_{s,y} k_s r_y} \leq 0 \tag{23}$$

is convex with respect to $k_s$ and $r_y$. Given two arbitrary feasible pairs of $k_s$ and $r_y$ satisfying Equation (23), namely $(k_s^a, r_y^a)$ and $(k_s^b, r_y^b)$, we only need to show that $(k_s^m, r_y^m)$ also satisfies Equation (23), where $k_s^m = (k_s^a + k_s^b)/2$, $r_y^m = (r_y^a + r_y^b)/2$. Indeed,

$$1 - \rho^2 - \sum_{s=1}^{S} \sum_{y=1}^{C} \sqrt{p_{s,y} k_s^m r_y^m}$$

$$= 1 - \rho^2 - \frac{1}{2} \sum_{s=1}^{S} \sum_{y=1}^{C} \sqrt{p_{s,y}} \cdot \sqrt{k_s^a + k_s^b} \cdot \sqrt{r_y^a + r_y^b}$$

$$\leq 1 - \rho^2 - \frac{1}{2} \sum_{s=1}^{S} \sum_{y=1}^{C} \sqrt{p_{s,y}} \cdot \left( \sqrt{k_s^a r_y^a} + \sqrt{k_s^b r_y^b} \right) \quad \text{(Cauchy's inequality)}$$

$$= \frac{1}{2} \left( 1 - \rho^2 \sum_{s=1}^{S} \sum_{y=1}^{C} \sqrt{p_{s,y} k_s^a r_y^a} \right) + \frac{1}{2} \left( 1 - \rho^2 \sum_{s=1}^{S} \sum_{y=1}^{C} \sqrt{p_{s,y} k_s^b r_y^b} \right)$$

$$\leq 0.$$

$\square$

## C.4 Proof of Lemma 3.1

*Proof of Lemma 3.1.* The proof of Lemma 3.1 is composed of two parts: (1) the optimization problem provides a fairness certificate for Problem 1; and (2) the certificate is tight. The high-level proof sketch is similar to the proof of Thm. 2.

**(1)** Suppose that the maximum of Problem 1 is attained with the test distribution $\mathcal{Q}^\star$ under the general shifting setting, then we decompose both $\mathcal{P}$ and $\mathcal{Q}^\star$ according to both the sensitive attribute and the label:

$$\mathcal{P} = \sum_{s=1}^{S}\sum_{y=1}^{C} p_{s,y}\mathcal{P}_{s,y}, \quad \mathcal{Q}^\star = \sum_{s=1}^{S}\sum_{y=1}^{C} q_{s,y}^\star \mathcal{Q}_{s,y}^\star. \tag{24}$$

Unlike sensitive shifting setting, in general shifting setting, here the subpopulation of $\mathcal{Q}^\star$ is $\mathcal{Q}_{s,y}^\star$ instead of $\mathcal{P}_{s,y}$ due to the existence of distribution shifting within each subpopulation.

Following the same argument as in the first part proof of Thm. 2, since $\mathcal{Q}^\star$ is a fair base rate distribution, we can define

$$k_s^\star := \Pr_{(X,Y)\sim\mathcal{Q}^\star}[X_s = s], \quad r_y^\star := \Pr_{(X,Y)\sim\mathcal{Q}^\star}[Y = y], \tag{25}$$

and write

$$\mathcal{Q}^\star := \sum_{s=1}^{S}\sum_{y=1}^{C} k_s^\star r_y^\star \mathcal{Q}_{s,y}^\star \tag{26}$$

since $q_{s,y}^\star = k_s^\star r_y^\star$. We also define $\rho_{s,y}^\star = H(\mathcal{P}_{s,y}, \mathcal{Q}_{s,y}^\star)$. Now we show these $k_s^\star, r_y^\star, \mathcal{Q}_{s,y}^\star, \rho_{s,y}^\star$ along with model parameter $\theta$ constitute a feasible point of Equation (6), and the objectives of Equation (6) and Problem 2 are the same given $\mathcal{Q}^\star$.

- (Feasibility)
  There are three constraints in Equation (6). By the definition of $k_s^\star$ and $r_y^\star$, naturally Equation (6b) is satisfied. Then, according to Equation (14) and the definifition of $\rho_{s,y}^\star$ above, Equation (6c) and Equation (6d) are satisfied.

- (Objective Equality)

$$
\begin{aligned}
\text{Equation (6a)} &= \sum_{s=1}^{S}\sum_{y=1}^{C} k_s^\star r_y^\star \mathbb{E}_{(X,Y)\sim\mathcal{Q}_{s,y}^\star}[\ell(h_\theta(X), Y)] \\
&= \sum_{s=1}^{S}\sum_{y=1}^{C} q_{s,y}^\star \mathbb{E}_{(X,Y)\sim\mathcal{Q}_{s,y}^\star}[\ell(h_\theta(X), Y)] \\
&= \mathbb{E}_{(X,Y)\sim\mathcal{Q}^\star}[\ell(h_\theta(X), Y)] = \text{Optimal value of Problem 1.}
\end{aligned} \tag{27}
$$

As a result, the optimal value of Equation (6) is larger than or equal to the optimal value of Problem 1, and hence the optimization problem encoded by Equation (6) provides a fairness certificate.

**(2)** To prove the tightness of the certificate, we only need to show that the optimal value of the optimization problem in Equation (6) is also attainable by the original Problem 1.

Suppose that the optimal objective of Equation (6) is achieved by optimizable parameters $k_s^\star, r_y^\star, \mathcal{Q}^\star$, and $\rho_{s,y}^\star$. Then, we construct $\mathcal{Q}^\dagger = \sum_{s=1}^{S}\sum_{y=1}^{C} k_s^\star r_y^\star \mathcal{Q}_{s,y}^\star$. We first show that $\mathcal{Q}^\dagger$ is a feasible point of Problem 1, and then show that the objective given $\mathcal{Q}^\dagger$ is equal to the optimal objective of Equation (6).

- (Feasibility)
  There are two constraints in Problem 1: the bounded distance constraint and the fair base rate constraint. The bounded distance constraint is satisfied due to applying Equation (14) along with Equations (6c) and (6d). The fair base rate constraint is satisfied following the same deduction as in Equation (21).

- (Objective Equality)

$$\text{Objective Problem 1} = \mathbb{E}_{(X,Y)\sim\mathcal{Q}^\dagger}[\ell(h_\theta(X),Y)] = \sum_{s=1}^{S}\sum_{y=1}^{C} k_s^\star r_y^\star \mathbb{E}_{(X,Y)\sim\mathcal{Q}_{s,y}^\star}[\ell(h_\theta(X),Y)]$$

$$= \text{Optimal value of Equation (6)}.$$

Thus, the optimal value of the optimization problem in Equation (6) is attainable also by the original Problem 1 which concludes the tightness proof.

$\square$

### C.5 Proof of Thm. 3

**High-Level Illustration.** The starting point of our proof is Lemma 3.1, where we have shown a fairness certificate for Problem 1 (general shifting setting). Then, we plug in Thm. 2.2 in [47] (stated as Thm. 4 in Appendix B.2) to upper bound the expected loss within each sub-population. Now, we get an optimization problem involving $k_s$, $r_y$, and $\rho_{s,y}$ that upper bounds the optimization problem in Lemma 3.1. In this optimization problem, we find $k_s$ and $r_y$ are bounded in $[0,1]$, and once these two variables are fixed, the optimization with respect to $x_{s,y} := (1 - \rho_{s,y}^2)^2$ becomes convex. Using this observation, we propose to partition the feasible space of $k_s$ and $r_y$ into sub-regions and solve the convex optimization within each region bearing some degree relaxation, which yields Thm. 3.

*Proof of Thm. 3.* The proof is done stage-wise: starting from Lemma 3.1, we apply relaxation and derive a subsequent optimization problem that upper bounds the previous one stage by stage, until we get the final expression in Thm. 3.

To demonstrate the proof, we first define the optimization problems at each stage, then prove the relaxations between each adjacent stage, and finally show that the last optimization problem contains a finite number of **C**'s values where each **C** is a convex optimization, so that the final optimization problem provides a computable fairness certificate.

We define these quantities, for $s \in [S], y \in [C]$:

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

$$(1 - \bar{\gamma}_{s,y}^2)^2 \le x_{s,y} \le 1 \quad \forall s \in [S], y \in [C]. \tag{33e}$$

We have this relation:

$$\text{Problem 1} \underbrace{\le}_{\text{Lemma 3.1}} (29) \overbrace{\underbrace{\le}_{\textbf{(A)}}}^{\substack{\text{(when } \ell(h_\theta(X), Y) \in [0, M] \\ \text{and } H(\mathcal{P}_{s,y}, \mathcal{Q}_{s,y}) \le \bar{\gamma}_{s,y})}} (30) \underbrace{=}_{\textbf{(B)}} (31) \underbrace{=}_{\textbf{(C)}} (32) \underbrace{\le}_{\textbf{(D)}} (33). \tag{34}$$

Thus, when $H(\mathcal{P}_{s,y}, \mathcal{Q}_{s,y}) \le \bar{\gamma}_{s,y}$ and $\sup_{(X,Y)\in\mathcal{X}\times\mathcal{Y}} \ell(h_\theta(X), Y) \le M$, given that $\ell$ is a non-negative loss by Section 2, we can see Equation (33), i.e., the expression in Thm. 3's statement, upper bounds Problem 1, i.e., provides a fairness certificate for Problem 1. The proofs of these equalities/inequalities are in the following parts labeled by **(A)**, **(B)**, **(C)**, and **(D)** respectively.

Now we show that each **C** queried by Equation (7) (or equally Equation (33a)) is a convex optimization. Inspecting **C**'s objective, with respect to the optimizable variable $x_{s,y}$, we find that the only non-linear term in the objective is $\sum_{s=1}^{S} \sum_{y=1}^{C} 2\overline{k_s}\overline{r_y}\sqrt{V_{s,y}}\sqrt{x_{s,y}(1 - x_{s,y})}$. Consider the function $f(x) = \sqrt{x(1 - x)}$. Define $g(y) = \sqrt{y}$ and $h(x) = x(1 - x)$, and then $f(x) = g(h(x))$. Thus, $f'(x) = g'(h(x))h'(x)$ and $f''(x) = g''(h(x))h'(x)^2 + g'(h(x))h''(x)$. Notice that $g''(h(x)) \le 0$, $g'(h(x)) > 0$, and $h''(x) < 0$ for $x \in (0, 1]$. Thus, $f''(x) \le 0$. Since $f$ is twice differentiable in $(0, 1]$, we can conclude that $f$ is concave and so does the objective of Equation (7). Inspecting **C**'s constraints, we observe that the only non-linear constraint is $\sum_{s=1}^{S} \sum_{y=1}^{C} \sqrt{p_{s,y}\overline{k_s}\overline{r_y}x_{s,y}} \ge 1 - \rho^2$. Due to the concavity of function $x \mapsto \sqrt{x}$, we have $\sqrt{p_{s,y}\overline{k_s}\overline{r_y}(x_{s,y}^a + x_{s,y}^b)/2} \ge \frac{1}{2}\left(\sqrt{p_{s,y}\overline{k_s}\overline{r_y}x_{s,y}^a} + \sqrt{p_{s,y}\overline{k_s}\overline{r_y}x_{s,y}^b}\right)$ for any two feasible points $x_{s,y}^a$ and $x_{s,y}^b$. Thus, this non-linear constraint defines a convex region. To this point, we have shown that **C**'s objective is concave and **C**'s constraints are convex, given that **C** is a maximization problem, **C** is a convex optimization. $\square$

Under the assumptions that $\ell(h_\theta(X), Y) \in [0, M]$ and $H(\mathcal{P}_{s,y}, \mathcal{Q}_{s,y}) \le \bar{\gamma}_{s,y}$:

**(A)** *Proof of Equation (29) $\le$ Equation (30).*
Given Equation (29d), for each $\mathcal{Q}_{s,y}$, applying Thm. 4, we get

$$\mathbb{E}_{(X,Y)\sim\mathcal{Q}_{s,y}}[\ell(h_\theta(X), Y)] \le E_{s,y} + 2\sqrt{\rho_{s,y}^2(1 - \rho_{s,y}^2)^2(2 - \rho_{s,y}^2)}\sqrt{V_{s,y}} + \rho_{s,y}^2(2 - \rho_{s,y}^2)C_{s,y}. \tag{35}$$

Plugging this inequality into all $\mathbb{E}_{(X,Y)\sim\mathcal{Q}_{s,y}}[\ell(h_\theta(X), Y)]$ in Equation (29a), we obtain Equation (30). $\square$

**(B)** *Proof of Equation (30) = Equation (31).*
By Equation (30d), $\rho_{s,y} \in [0, 1]$. Therefore, $x_{s,y} := (1 - \rho_{s,y}^2)^2$ is a one-to-one mapping, and we can use $x_{s,y}$ to parameterize $\rho_{s,y}$, which yields Equation (31). $\square$

**(C)** *Proof of Equation (31) = Equation (32).*
From Equation (31b), we notice that the feasible range of $k_s$ and $r_y$ is subsumed by $[0, 1]$. We now partition this region $[0, 1]$ for each variable to $T$ sub-regions: $[(i-1)/T, i/T]$, $i \in [T]$, and then consider the maximum value across all the combinations of each sub-region for variables $k_s$ and $r_y$, when feasible. As a result, Equation (31) can be written as the maximum over all such sub-problems where $k_s$'s and $r_y$'s enumerate all possible sub-region combinations, which is exactly encoded by Equation (32). $\square$

**(D)** *Proof of Equation (32) $\le$ Equation (33).*
We only need to show that when $\mathbf{C}'\left(\{[\underline{k_s}, \overline{k_s}]\}_{s=1}^{S}, \{[\underline{r_y}, \overline{r_y}]\}_{y=1}^{C}\right)$ is feasible,

$$\mathbf{C}'\left(\{[\underline{k_s}, \overline{k_s}]\}_{s=1}^{S}, \{[\underline{r_y}, \overline{r_y}]\}_{y=1}^{C}\right) \le \mathbf{C}\left(\{[\underline{k_s}, \overline{k_s}]\}_{s=1}^{S}, \{[\underline{r_y}, \overline{r_y}]\}_{y=1}^{C}\right). \tag{36}$$

Since both $\mathbf{C}'$ and $\mathbf{C}$ are maximization problem, we only need to show that the objective of $\mathbf{C}$ upper bounds that of $\mathbf{C}'$, and the constraints of $\mathbf{C}'$ are equal or relaxations of those of $\mathbf{C}$.

For the objective, given that $\underline{k_s} \leq k_s \leq \overline{k_s}$ and $\underline{r_y} \leq r_y \leq \overline{r_y}$, for any $x_{s,y}$, We observe that

$$k_s r_y (E_{s,y} + C_{s,y}) \leq \overline{k_s}\overline{r_y} (E_{s,y} + C_{s,y})_+ + \underline{k_s}\underline{r_y} (E_{s,y} + C_{s,y})_- ,$$

$$k_s r_y \cdot \left( 2\sqrt{x_{s,y}(1 - x_{s,y})}\sqrt{V_{s,y}} \right) \leq \overline{k_s}\overline{r_y} \cdot \left( 2\sqrt{x_{s,y}(1 - x_{s,y})}\sqrt{V_{s,y}} \right), \tag{37}$$

$$-k_s r_y C_{s,y} x_{s,y} \leq -\underline{k_s}\underline{r_y} x_{s,y}(C_{s,y})_+ - \overline{k_s}\overline{r_y} x_{s,y}(C_{s,y})_-,$$

and by summing up all these terms for all $s \in [S]$ and $y \in [C]$, the LHS would be the objective of $\mathbf{C}'$ and the RHS would be the objective of $\mathbf{C}$. Hence, $\mathbf{C}$'s objective upper bounds that of $\mathbf{C}'$. For the constraints, similarly, given that $\underline{k_s} \leq k_s \leq \overline{k_s}$ and $\underline{r_y} \leq r_y \leq \overline{r_y}$, we have

$$(32c) \sum_{s=1}^{S} k_s = 1, \sum_{y=1}^{C} r_y = 1 \implies \sum_{s=1}^{S} \underline{k_s} \leq 1, \sum_{s=1}^{S} \overline{k_s} \geq 1, \sum_{y=1}^{C} \underline{r_y} \leq 1, \sum_{y=1}^{C} \overline{r_y} \geq 1 (33c),$$

$$(32d) \sum_{s=1}^{S} \sum_{y=1}^{C} \sqrt{p_{s,y} k_s r_y x_{s,y}} \geq 1 - \rho^2 \implies \sum_{s=1}^{S} \sum_{y=1}^{C} \sqrt{p_{s,y} \overline{k_s}\overline{r_y} x_{s,y}} \geq 1 - \rho^2 (33d),$$

$$(32e) \text{ is as same as } (33e),$$

which implies that all feasible solutions of $\mathbf{C}'$ are also feasible for $\mathbf{C}$. Combining with the fact that for any solution of $\mathbf{C}'$, its objective value $\mathbf{C}$ is greater than or equal to that of $\mathbf{C}'$ as shown above, we have Equation (36) which concludes the proof. $\qquad\square$

# D  Omitted Theorem Statements and Proofs for Finite Sampling Error

## D.1  Finite Sampling Confidence Intervals

**Lemma D.1.** *Let $\hat{P}$ be set of i.i.d. finite samples from $\mathcal{P}$, and let $\hat{P}_{s,y} := \{(X_i, Y_i) \in \hat{P} : (X_i)_s = s, Y_i = y\}$ for any $s \in [S], y \in [C]$. Let $\ell : (\hat{y}, y) \to [0, M]$ be a loss function. We define $\hat{L}_n = \frac{1}{|\hat{P}_{s,y}|} \sum_{(X_i,Y_i)\in \hat{P}_{s,y}} \ell(h_\theta(X_i), Y_i), s_n^2 = \frac{1}{n(n-1)} \sum_{1\leq i<j\leq n}^{n} (\ell(h_\theta(X_i), Y) - \ell(h_\theta(X_j), Y))^2$, and $\hat{P}_{s,y} := \{(X_i, Y_i) \in \hat{P} : (X_i)_s = s, Y_i = y\}$. Then for $\delta > 0$, with respect to the random draw of $\hat{P}$ from $\mathcal{P}$, we have*

$$\Pr\left( \hat{L}_n - M\sqrt{\frac{\ln(2/\delta)}{2|\hat{P}_{s,y}|}} \leq \mathop{\mathbb{E}}_{(X,Y)\sim\mathcal{P}_{s,y}} [\ell(h_\theta(X), Y)] \leq \hat{L}_n + M\sqrt{\frac{\ln(2/\delta)}{2|\hat{P}_{s,y}|}} \right) \geq 1 - \delta, \tag{38}$$

$$\Pr\left( \sqrt{s_n^2} - M\sqrt{\frac{2\ln(2/\delta)}{|\hat{P}_{s,y}| - 1}} \leq \sqrt{\mathop{\mathbb{V}}_{(X,Y)\sim\mathcal{P}_{s,y}} [\ell(h_\theta(X), Y)]} \leq \sqrt{s_n^2} + M\sqrt{\frac{2\ln(2/\delta)}{|\hat{P}_{s,y}| - 1}} \right) \geq 1 - \delta, \tag{39}$$

$$\Pr\left( \frac{|\hat{P}_{s,y}|}{|\hat{P}|} - \sqrt{\frac{\ln(2/\delta)}{2|\hat{P}|}} \leq \mathop{\Pr}_{(X,Y)\sim\mathcal{P}} [X_s = s, Y = y] \leq \frac{|\hat{P}_{s,y}|}{|\hat{P}|} + \sqrt{\frac{\ln(2/\delta)}{2|\hat{P}|}} \right) \geq 1 - \delta. \tag{40}$$

*Proof of Lemma D.1.* We can get Equation (39) according to Theorem 10 in [32]. Here, we will provide proofs for Equation (38) and Equation (40), respectively. The general idea is to use Hoeffding's inequality to get the high-confidence interval.

We will prove Equation (38) first. From Hoeffding's inequality, for all $t > 0$, we have:

$$\Pr\left( \left| \hat{L}_n - \mathop{\mathbb{E}}_{(X,Y)\sim\mathcal{P}_{s,y}} [\ell(h_\theta(X), Y)] \right| \geq t \right) \leq 2\exp\left( -\frac{2|\hat{P}_{s,y}|^2 t^2}{|\hat{P}_{s,y}| M^2} \right) \tag{41}$$

Since we want to get an interval with confidence $1 - \delta$, we let $2\exp\left( -\frac{2|\hat{P}_{s,y}|^2 t^2}{|\hat{P}_{s,y}| M^2} \right) = \delta$, from which we can derive that

$$t = M\sqrt{\frac{\ln(2/\delta)}{2|\hat{P}_{s,y}|}} \tag{42}$$

Plugging Equation (42) into Equation (41), we can get:

$$\Pr\left(\hat{L}_n - M\sqrt{\frac{\ln(2/\delta)}{2|\hat{P}_{s,y}|}} \leq \mathop{\mathbb{E}}_{(X,Y)\sim\mathcal{P}_{s,y}}[\ell(h_\theta(X),Y)] \leq \hat{L}_n + M\sqrt{\frac{\ln(2/\delta)}{2|\hat{P}_{s,y}|}}\right) \geq 1-\delta \quad (43)$$

Then we will prove Equation (40). From Hoeffding's inequality, for all $t > 0$, we have:

$$\Pr\left(\left|\frac{|\hat{P}_{s,y}|}{|\hat{P}|} - \mathop{\Pr}_{(X,Y)\sim\mathcal{P}}[X_s = s, Y = y]\right| \geq t\right) \leq 2\exp\left(-\frac{2|\hat{P}|^2 t^2}{|\hat{P}|}\right) \quad (44)$$

Since we want to get an interval with confidence $1-\delta$, we let $2\exp\left(-\dfrac{2|\hat{P}|^2 t^2}{|\hat{P}|}\right) = \delta$, from which we can derive that

$$t = \sqrt{\frac{\ln(2/\delta)}{2|\hat{P}|}} \quad (45)$$

Plugging Equation (45) into Equation (44), we can get:

$$\Pr\left(\frac{|\hat{P}_{s,y}|}{|\hat{P}|} - \sqrt{\frac{\ln(2/\delta)}{2|\hat{P}|}} \leq \mathop{\Pr}_{(X,Y)\sim\mathcal{P}}[X_s = s, Y = y] \leq \frac{|\hat{P}_{s,y}|}{|\hat{P}|} + \sqrt{\frac{\ln(2/\delta)}{2|\hat{P}|}}\right) \geq 1-\delta \quad (46)$$

$\square$

### D.2 Fairness Certification Statements with Finite Sampling

**Theorem 5** (Thm. 2 with finite sampling). *Given a distance bound $\rho > 0$ and any $\delta > 0$, the following constrained optimization, which is **convex**, when feasible, provides a fairness certificate for Problem 2 with probability at least $1 - 2SC\delta$:*

$$\max_{k_s, r_y, p_{s,y}} \quad \sum_{s=1}^{S}\sum_{y=1}^{C} k_s r_y \overline{E_{s,y}} \quad (47a)$$

$$\text{s.t.} \quad \sum_{s=1}^{S} k_s = 1, \quad \sum_{y=1}^{C} r_y = 1, \quad k_s \geq 0 \quad \forall s \in [S], \quad r_y \geq 0 \quad \forall y \in [C], \quad (47b)$$

$$1 - \rho^2 - \sum_{s=1}^{S}\sum_{y=1}^{C}\sqrt{p_{s,y} k_s r_y} \leq 0, \quad (47c)$$

$$\underline{p_{s,y}} \leq p_{s,y} \leq \overline{p_{s,y}}, \quad \forall s \in [S], \quad \forall y \in [C] \quad (47d)$$

$$\sum_{s=1}^{S}\sum_{y=1}^{C} p_{s,y} = 1 \quad (47e)$$

*where $\overline{E_{s,y}} := \hat{L}_n + M\sqrt{\ln(2/\delta)/\left(2|\hat{P}_{s,y}|\right)}$, $\underline{p_{s,y}} := |\hat{P}_{s,y}|/|\hat{P}| - \sqrt{\ln(2/\delta)/\left(2|\hat{P}|\right)}$, $\overline{p_{s,y}} := |\hat{P}_{s,y}|/|\hat{P}| + \sqrt{\ln(2/\delta)/\left(2|\hat{P}|\right)}$ are constants computed with Lemma D.1.*

**Theorem 6.** *If for any $s \in [S]$ and $y \in [Y]$, $H(\mathcal{P}_{s,y}, \mathcal{Q}_{s,y}) \leq \bar{\gamma}_{s,y}$ and $0 \leq \sup_{(X,Y)\in\mathcal{X}\times\mathcal{Y}} \ell(h_\theta(X), Y) \leq M$, given a distance bound $\rho > 0$ and any $\delta > 0$, for any region granularity $T \in \mathbb{N}_+$, the following expression provides a fairness certificate for Problem 1 with probability at least $1 - 3SC\delta$:*

$$\bar{\ell} = \max_{\{i_s \in [T]: s \in [S]\}, \{j_y \in [T]: y \in [C]\}} \mathbf{C}\left(\left\{\left[\frac{i_s - 1}{T}, \frac{i_s}{T}\right]\right\}_{s=1}^{S}, \left\{\left[\frac{j_y - 1}{T}, \frac{j_y}{T}\right]\right\}_{y=1}^{C}\right), \text{ where} \quad (48)$$

$$\mathbf{C}\left(\{[\underline{k_s}, \overline{k_s}]\}_{s=1}^{S}, \{[\underline{r_y}, \overline{r_y}]\}_{y=1}^{C}\right) = \max_{x_{s,y}, p_{s,y}} \sum_{s=1}^{S}\sum_{y=1}^{C}\left(\overline{k_s}\overline{r_y}\left(\overline{E_{s,y}} + \overline{C_{s,y}}\right)_+ + \underline{k_s}\underline{r_y}\left(\overline{E_{s,y}} + \overline{C_{s,y}}\right)_-\right.$$

$$+2\overline{k_s}\overline{r_y}\sqrt{x_{s,y}(1-x_{s,y})}\sqrt{\overline{V_{s,y}}} - \underline{k_s}\underline{r_y}x_{s,y}(\underline{C_{s,y}})_+ - \overline{k_s}\overline{r_y}x_{s,y}(\overline{C_{s,y}})_-\Bigg) \tag{49a}$$

$$\text{s.t.} \quad \sum_{s=1}^{S}\underline{k_s} \le 1, \quad \sum_{s=1}^{S}\overline{k_s} \ge 1, \quad \sum_{y=1}^{C}\underline{r_y} \le 1, \quad \sum_{y=1}^{C}\overline{r_y} \ge 1, \tag{49b}$$

$$\sum_{s=1}^{S}\sum_{y=1}^{C}\sqrt{p_{s,y}\overline{k_s}\overline{r_y}x_{s,y}} \ge 1-\rho^2, \quad \left(1-\overline{\overline{\gamma}_{s,y}^2}\right)^2 \le x_{s,y} \le 1, \tag{49c}$$

$$\underline{p_{s,y}} \le p_{s,y} \le \overline{p_{s,y}}, \quad \sum_{s=1}^{S}\sum_{y=1}^{C}p_{s,y} = 1 \tag{49d}$$

*where* $(\cdot)_+ = \max\{\cdot,0\}$, $(\cdot)_- = \min\{\cdot,0\}$; $\underline{E_{s,y}} := \hat{L}_n - M\sqrt{\ln(2/\delta)/\left(2|\hat{P}_{s,y}|\right)}$, $\overline{E_{s,y}} := \hat{L}_n + M\sqrt{\ln(2/\delta)/\left(2|\hat{P}_{s,y}|\right)}$, $\underline{V_{s,y}} = \left(\sqrt{s_n^2} - M\sqrt{2\ln(2/\delta)/\left(|\hat{P}_{s,y}|-1\right)}\right)^2$, $\overline{V_{s,y}} = \left(\sqrt{s_n^2} + M\sqrt{2\ln(2/\delta)/\left(|\hat{P}_{s,y}|-1\right)}\right)^2$, $\underline{p_{s,y}} := |\hat{P}_{s,y}|/|\hat{P}| - \sqrt{\ln(2/\delta)/\left(2|\hat{P}|\right)}$, $\overline{p_{s,y}} := |\hat{P}_{s,y}|/|\hat{P}|+\sqrt{\ln(2/\delta)/\left(2|\hat{P}|\right)}$ *computed with Lemma* D.1, *and* $\underline{C_{s,y}} = M-\overline{E_{s,y}}-V_{s,y}/(M-\overline{E_{s,y}})$, $\overline{C_{s,y}} = M - \underline{E_{s,y}} - V_{s,y}/(M - \underline{E_{s,y}})$, $\overline{\overline{\gamma}_{s,y}^2} = 1 - (1 + (M - \underline{E_{s,y}})^2/\overline{V_{s,y}})^{-\frac{1}{2}}$. *Equation* (48) *only takes* **C**'s *value when it is feasible, and each* **C** *queried by Equation* (48) *is a **convex optimization**.*

### D.3 Proofs of Fairness Certification with Finite Sampling

**High-Level Illustration.** We use Hoeffding's inequality to bound the finite sampling error of statistics and add the high confidence box constraints to the optimization problems, which can still be proved to be convex.

*Proof of Thm.* 5. The proof of Thm. 5 is composed of two parts: (1) the optimization problem provides a fairness certificate for Problem 2; (2) the optimization problem is convex.

**(1)** We prove that Thm. 5 provides a fairness certificate for Problem 2 in this part. Since Thm. 2 provides a fairness certificate for Problem 2, we only need to prove: (a) the feasible region of the optimization problem in Thm. 2 is a subset of the feasible region of the optimization problem in Thm. 5, and (b) the optimization objective in Thm. 2 can be upper bounded by that in Thm. 5.

To prove (a), we first equivalently transform the optimization problem in Thm. 2 into the following optimization problem by adding $p_{sy}$ to the decision variables:

$$\max_{k_s,r_y,p_{s,y}} \quad \sum_{s=1}^{S}\sum_{y=1}^{C}k_s r_y E_{s,y} \tag{50a}$$

$$\text{s.t.} \quad \sum_{s=1}^{S}k_s = 1, \quad \sum_{y=1}^{C}r_y = 1, \quad k_s \ge 0 \quad \forall s \in [S], \quad r_y \ge 0 \quad \forall y \in [C], \tag{50b}$$

$$1 - \rho^2 - \sum_{s=1}^{S}\sum_{y=1}^{C}\sqrt{p_{s,y}k_s r_y} \le 0, \tag{50c}$$

$$p_{s,y} = |\hat{P}_{s,y}|/|\hat{P}|, \quad \forall s \in [S], \quad \forall y \in [C] \tag{50d}$$

$$\sum_{s=1}^{S}\sum_{y=1}^{C}p_{s,y} = 1 \tag{50e}$$

For decision variables $k_{s,y}$ and $r_{s,y}$, optimization 47 and 50 has the same constraints (Equation (47b) and Equation (50b)). For decision variables $p_{s,y}$, the feasible region of $p_{s,y}$ in

optimization 47 (decided by Equations (47d) and (47e)) is a subset of the feasible region of $p_{s,y}$ in optimization 50 (decided by Equations (50d) and (50e)), since $p_{s,y} \leq |\hat{P}_{s,y}|/|\hat{P}| \leq \overline{p_{s,y}}$. Therefore, the feasible region with respect to $k_{s,y}$, $r_{s,y}$, and $p_{s,y}$ of the optimization problem in Thm. 2 is a subset of that in Thm. 5.

To prove (b), we only need to show that the objective in Equation (47a) can be upper bounded by the objective in Equation (50a). The statement $\sum_{s=1}^{S}\sum_{y=1}^{C} k_s r_y E_{s,y} \leq \sum_{s=1}^{S}\sum_{y=1}^{C} k_s r_y \overline{E_{s,y}}$ consistently holds because $E_{s,y} \leq \overline{E_{s,y}}$ and $k_s, r_y \geq 0$.

Combining the proofs of (a) and (b), we prove that Thm. 5 provides a fairness certificate for Problem 2.

**(2)** Inspecting that the objective and all the constraints in optimization problem in Equation (47) are linear with respect to $k_s$, $r_y$, $p_{s,y}$ but the one in Equation (47c). Therefore, we only need to prove that the following constraint is convex with respect to $k_s$, $r_y$, $p_{s,y}$:

$$1 - \rho^2 - \sum_{s=1}^{S}\sum_{y=1}^{C} \sqrt{p_{s,y} k_s r_y} \leq 0 \tag{51}$$

We define a function $f$ with respect to vector $\mathbf{p} := [p_{s,y}]_{s \in [S], y \in [C]}$: $f(\mathbf{p}) = 1 - \rho^2 - \sum_{s=1}^{S}\sum_{y=1}^{C} \sqrt{p_{s,y} k_s r_y}$. Then we can derive that:

$$\frac{\partial^2 f}{\partial \mathbf{p}^2} = \sum_{s=1}^{S}\sum_{y=1}^{C} \frac{\sqrt{k_s r_y}}{4} p_{s,y}^{-\frac{3}{2}} \geq 0 \tag{52}$$

Therefore, the function $f$ is convex with respect to $p_{s,y}$. Similarly, we can prove the convexity with respect to $k_s$ and $r_y$. Finally, we can conclude that the constraint in Equation (51) is convex with respect to $k_s$, $r_y$, $p_{s,y}$ and the optimization problem defined in Thm. 5 is convex.

Since we use the union bound to bound $E_{s,y}$ and $p_{s,y}$ for all $s \in [S], y \in [C]$ simultaneously, the confidence is $1 - 2SC\delta$. □

*Proof of Thm. 6.* The proof of Thm. 6 includes two parts: (1) the optimization problem provides a fairness certificate for Problem 1; (2) each **C** queried by Equation (48) is a convex optimization.

**(1)** Since Thm. 3 provides a fairness certificate for Problem 1, we only need to prove: (a) the feasible region of the optimization problem in Thm. 3 is a subset of that in Thm. 6, and (b) the optimization objective in Thm. 3 can be upper bounded by that in Thm. 6.

To prove (a), we first equivalently transform the optimization problem in Thm. 3 into the following optimization problem by adding $p_{sy}$ to the decision variables:

$$\mathbf{C}\left(\{[\underline{k_s}, \overline{k_s}]\}_{s=1}^{S}, \{[\underline{r_y}, \overline{r_y}]\}_{y=1}^{C}\right) = \max_{x_{s,y}, p_{s,y}} \sum_{s=1}^{S}\sum_{y=1}^{C} \left(\overline{k_s r_y}\left(E_{s,y} + C_{s,y}\right)_+ + \underline{k_s r_y}\left(E_{s,y} + C_{s,y}\right)_- \right.$$

$$\left. + 2\overline{k_s r_y}\sqrt{x_{s,y}(1 - x_{s,y})}\sqrt{V_{s,y}} - \underline{k_s r_y} x_{s,y}(C_{s,y})_+ - \overline{k_s r_y} x_{s,y}(C_{s,y})_- \right) \tag{53a}$$

$$\text{s.t. } \sum_{s=1}^{S} \underline{k_s} \leq 1, \quad \sum_{s=1}^{S} \overline{k_s} \geq 1, \sum_{y=1}^{C} \underline{r_y} \leq 1, \quad \sum_{y=1}^{C} \overline{r_y} \geq 1, \tag{53b}$$

$$\sum_{s=1}^{S}\sum_{y=1}^{C} \sqrt{p_{s,y}\overline{k_s r_y} x_{s,y}} \geq 1 - \rho^2, \quad \left(1 - \bar{\gamma}_{s,y}^2\right)^2 \leq x_{s,y} \leq 1, \tag{53c}$$

$$p_{s,y} = |\hat{P}_{s,y}|/|\hat{P}|, \quad \forall s \in [S], \quad \forall y \in [C] \tag{53d}$$

$$\sum_{s=1}^{S}\sum_{y=1}^{C} p_{s,y} = 1 \tag{53e}$$

For decision varibales $x_{s,y}$, since $\sqrt{p_{s,y}\underline{k_s}\underline{r_y}x_{s,y}} \leq \sqrt{\overline{p_{s,y}k_s r_y}x_{s,y}}$ and $\left(1-\bar{\gamma}_{s,y}^2\right)^2 \geq \left(1-\overline{\bar{\gamma}_{s,y}^2}\right)^2$, the feasible region of $x_{s,y}$ in Equation (53) is a subset of that in Equation (49). For decision variables $p_{s,y}$, since $\underline{p_{s,y}} \leq |\hat{P}_{s,y}|/|\hat{P}| \leq \overline{p_{s,y}}$, the feasible region of $p_{s,y}$ in Equation (53) is also a subset of that in Equation (49). Therefore, the feasible region of the optimization problem in Thm. 3 is a subset of that in Thm. 6.

To prove (b), we only need to show that the objective in Equation (53a) can be upper bounded by the objective in Equation (49a). Since $\underline{k_s}, \overline{k_s}, \underline{r_y}, \overline{r_y} \geq 0$ and $0 \leq x_{s,y} \leq 1$ hold, we can observe that $\forall s \in [S], \forall y \in [C]$,

$$\overline{k_s r_y}\left(E_{s,y}+C_{s,y}\right)_+ + \underline{k_s}\underline{r_y}\left(E_{s,y}+C_{s,y}\right)_- + 2\overline{k_s r_y}\sqrt{x_{s,y}(1-x_{s,y})}\sqrt{V_{s,y}} - \underline{k_s}\underline{r_y}x_{s,y}(C_{s,y})_+ -$$

$$\overline{k_s r_y}x_{s,y}(C_{s,y})_- \leq \overline{k_s r_y}\left(\overline{E_{s,y}}+\overline{C_{s,y}}\right)_+ + \underline{k_s}\underline{r_y}\left(\overline{E_{s,y}}+\overline{C_{s,y}}\right)_- + 2\overline{k_s r_y}\sqrt{x_{s,y}(1-x_{s,y})}\sqrt{V_{s,y}}$$

$$- \underline{k_s}\underline{r_y}x_{s,y}(\overline{C_{s,y}})_+ - \overline{k_s r_y}x_{s,y}(\overline{C_{s,y}})_-$$

Therefore, we prove that the optimization in Thm. 6 provides a fairness certificate for Problem 1.

(2) We will prove that each **C** queried by Equation (48) is a convex optimization with respect to decision variables $x_{s,y}$ and $p_{s,y}$ in this part. In the proof of Thm. 3, we provide the proof of convexity with respect to $x_{s,y}$, so we only need to prove that the optimization problem is convex with respect to $p_{s,y}$. We can observe that the constraints of $p_{s,y}$ in Equation (49d) is linear, and thus we only need to prove that $\sum_{s=1}^{S}\sum_{y=1}^{C}\sqrt{p_{s,y}\underline{k_s}\underline{r_y}x_{s,y}} \geq 1 - \rho^2$ (the constraint in Equation (49c)) is convex with respect to $p_{s,y}$. Here, we define a function $f$ with respect to vector $\boldsymbol{p} := [p_{s,y}]_{s \in [S], y \in [C]}$: $f(\boldsymbol{p}) = 1 - \rho^2 - \sum_{s=1}^{S}\sum_{y=1}^{C}\sqrt{p_{s,y}\underline{k_s}\underline{r_y}}$. Then we can derive that:

$$\left(\frac{\partial^2 f}{\partial \boldsymbol{p}^2}\right)_{sy,s'y'} = \sum_{s=1}^{S}\sum_{y=1}^{C}\frac{\sqrt{\underline{k_s}\underline{r_y}}}{4}p_{s,y}^{-\frac{3}{2}} \cdot \mathbb{I}[s=s', y=y'] \geq 0 \tag{55}$$

Thus, the function $f$ is convex and $f(\boldsymbol{p}) \leq 0$ defines a convex set with respect to $p_{s,y}$. Then, we prove that the constraint $\sum_{s=1}^{S}\sum_{y=1}^{C}\sqrt{p_{s,y}\underline{k_s}\underline{r_y}x_{s,y}} \geq 1 - \rho^2$ is convex with respect to $p_{s,y}$.

Since we use the union bound to bound $E_{s,y}, V_{s,y}$ and $p_{s,y}$ for all $s \in [S], y \in [C]$ simultaneously, the confidence is $1 - 3SC\delta$. □

# E  Experiments

## E.1  Datasets

We validate our certified fairness on *six* real-world datasets: Adult [3], Compas [2], Health [19], Lawschool [48], Crime [3], and German [3]. All the used datasets contain categorical data.

In Adult dataset, we have 14 attributes of a person as input and try to predict whether the income of the person is over 50k $/year. The sensitive attribute in Adult is selected as the sex.

In Compas dataset, given the attributes of a criminal defendent, the task is to predict whether he/she will re-offend in two years. The sensitive attribute in Compas is selected as the race.

In Health dataset, given the physician records and and insurance claims of the patients, we try to predict ten-year mortality by binarizing the Charlson Index, taking the median value as a cutoff. The sensitive attribute in Health is selected as the age.

In Lawschool dataset, we try to predict whether a student passes the exam according to the appication records of different law schools. The sensitive attribute in Lawschool is the race.

In Crime dataset, we try to predict whether a specific community is above or below the median number of violent crimes per population. The sensitive attribute in Crime is selected as the race.

In German dataset, each person is classified as good or bad credit risks according to the set of attributes. The sensitive attribute in German is selected as the sex.

Following [39], we consider the scenario where sensitive attributes and labels take binary values, and we also follow their standard data processing steps: (1) normalize the numerical values of all attributes with the mean value 0 and variance 1, (2) use one-hot encodings to represent categorical attributes, (3) drop instances and attributes with missing values, and (4) split the datasets into training set, validation set, and test set.

**Training Details.** We directly train a ReLU network composed of two hidden layers on the training set of six datasets respectively following the setting in [39]. Concretely, we train the models for 100 epochs with a batch size of 256. We adopt the binary cross-entropy loss and use the Adam optimizer with weight decay 0.01 and dynamic learning rate scheduling (ReduceLROnPlateau in [35]) based on the loss on the validation set starting at 0.01 with the patience of 5 epochs.

### E.2 Fair Base Rate Distribution Generation Protocol

To evaluate how well our certificates capture the fairness risk in practice, we compare our certification bound with the empirical loss evaluated on randomly generated $30,000$ fairness constrained distributions $\mathcal{Q}$ shifted from $\mathcal{P}$. Now, we introduce the protocols to generate fairness distributions $\mathcal{Q}$ for sensitive shifting and general shifting, respectively. Note that the protocols are only valid when the sensitive attributes and labels take binary values.

Fair base rate distributions generation steps in the *sensitive shifting* scenario:

**(1)** Sample the proportions of subpopulations of the generated distribution $q_{0,0}, q_{0,1}, q_{1,0}, q_{1,1}$: uniformly sample two real values in the interval $[0, 1]$, and do the assignment: $q_{0,0} := kr$, $q_{0,1} := k(1 - r)$, $q_{1,0} := (1 - k)r$, $q_{1,1} := (1 - k)(1 - r)$.

**(2)** Determine the sample size of every subpopulation: first determine the subpopulation which requires the largest sample size, use all the samples in that subpopulation, and then calculate the sample size in other subpopulations according to the proportions.

**(3)** Uniformly sample in each subpopulation based on the sample size.

**(4)** Calculate the Hellinger distance $H(\mathcal{P}, \mathcal{Q}) = \sqrt{1 - \sum_{s=0}^{1} \sum_{y=0}^{1} \sqrt{p_{s,y}} \sqrt{q_{s,y}}}$. Suppose that the support of $\mathcal{P}$ and $\mathcal{Q}$ is $\mathcal{X} \times \mathcal{Y}$ and the densities of $\mathcal{P}$ and $\mathcal{Q}$ with respect to a suitable measure are $f_{\mathcal{P}}$ and $f_{\mathcal{Q}}$, respectively. Since we consider sensitive shifting here, we have $f_{\mathcal{Q}_{s,y}} = \lambda_{s,y} f_{\mathcal{P}_{s,y}}$, $s, y \in \{0, 1\}$ where $\lambda_{s,y}$ is a scalor. The derivation of the distance calculation formula is shown as follows,

$$H^2(\mathcal{P}, \mathcal{Q}) = 1 - \iint_{\mathcal{X} \times \mathcal{Y}} \sqrt{f_{\mathcal{P}}(x,y)} \sqrt{f_{\mathcal{Q}}(x,y)} \mathrm{d}x \mathrm{d}y \tag{56a}$$

$$= 1 - \sum_{s=0}^{1} \sum_{y=0}^{1} \iint_{f_{\mathcal{P}_{s,y}}(x,y)>0} \sqrt{f_{\mathcal{P}_{s,y}}(x,y)} \sqrt{\lambda_{s,y} f_{\mathcal{P}_{s,y}}} \mathrm{d}x \mathrm{d}y \tag{56b}$$

$$= 1 - \sum_{s=0}^{1} \sum_{y=0}^{1} \sqrt{\lambda_{s,y}} \iint_{f_{\mathcal{P}_{s,y}}(x,y)>0} f_{\mathcal{P}_{s,y}}(x,y) \mathrm{d}x \mathrm{d}y \tag{56c}$$

$$= 1 - \sum_{s=0}^{1} \sum_{y=0}^{1} \sqrt{\lambda_{s,y}} p_{s,y} \tag{56d}$$

$$= 1 - \sum_{s=0}^{1} \sum_{y=0}^{1} \sqrt{p_{s,y}} \sqrt{q_{s,y}}. \tag{56e}$$

Fair base rate distribution generation steps in the *general shifting* scenario:

**(1)** Construct a data distribution $\mathcal{Q}'$ that is disjoint with the training data distribution $\mathcal{P}$ by changing the distribution of non-sensitive values given the sensitive attributes and labels.

(2) Sample mixing parameters $\alpha_{s,y}$ and $\alpha'_{s,y}$ in the interval $[0,1]$ satisfying $\frac{p_{00}\alpha_{00}+q_{00}\alpha'_{00}}{p_{01}\alpha_{01}+q_{01}\alpha'_{01}} = \frac{p_{10}\alpha_{10}+q_{10}\alpha'_{10}}{p_{11}\alpha_{11}+q_{11}\alpha'_{11}}$ (base rate parity) and $p_{00}\alpha_{00} + q_{00}\alpha'_{00} + p_{01}\alpha_{01} + q_{01}\alpha'_{01} + p_{10}\alpha_{10} + q_{10}\alpha'_{10} + p_{11}\alpha_{11} + q_{11}\alpha'_{11} = 1$.

(3) Determine the proportion of every subpopulation in distribution $\mathcal{Q}$: $q_{s,y} := \alpha_{s,y}p_{s,y} + \alpha'_{s,y}q'_{s,y}, \quad s, y \in \{0, 1\}$.

(4) Determine the sample size of every subpopulation in $\mathcal{P}$ and $\mathcal{Q}'$: first determine the subpopulation which requires the largest sample size, use all the samples in that subpopulation, and then calculate the sample size in other subpopulations according to the proportions.

(5) Calculate the Hellinger distance between distribution $\mathcal{P}$ and $\mathcal{Q}$: $H(\mathcal{P}, \mathcal{Q}) = \sqrt{1 - \sum_{s=0}^{1}\sum_{y=0}^{1}\sqrt{\alpha_{s,y}p_{s,y}}}$. Suppose that the support of $\mathcal{P}$ and $\mathcal{Q}$ is $\mathcal{X} \times \mathcal{Y}$ and the densities of $\mathcal{P}$ and $\mathcal{Q}$ with respect to a suitable measure are $f_{\mathcal{P}}$ and $f_{\mathcal{Q}}$, respectively. The derivation of the distance calculation formula is shown as follows,

$$H^2(\mathcal{P}, \mathcal{Q}) = 1 - \iint_{\mathcal{X}\times\mathcal{Y}} \sqrt{f_{\mathcal{P}}(x,y)}\sqrt{f_{\mathcal{Q}}(x,y)}\mathrm{d}x\mathrm{d}y \tag{57a}$$

$$= 1 - \iint_{\mathcal{X}\times\mathcal{Y}} \sqrt{\sum_{s=0}^{1}\sum_{y=0}^{1} f_{\mathcal{P}_{s,y}}(x,y)}\sqrt{\sum_{s=0}^{1}\sum_{y=0}^{1}\left(\alpha_{s,y}f_{\mathcal{P}_{s,y}}(x,y) + \alpha'_{s,y}f_{\mathcal{Q}'_{s,y}}(x,y)\right)}\mathrm{d}x\mathrm{d}y \tag{57b}$$

$$= 1 - \sum_{s=0}^{1}\sum_{y=0}^{1}\sqrt{\alpha_{s,y}}\iint_{f_{\mathcal{P}_{s,y}}(x,y)>0} f_{\mathcal{P}_{s,y}}(x,y)\sqrt{1 + \frac{\alpha'_{s,y}f_{\mathcal{Q}_{s,y}}(x,y)}{\alpha_{s,y}f_{\mathcal{P}_{s,y}}(x,y)}}\mathrm{d}x\mathrm{d}y \tag{57c}$$

$$= 1 - \sum_{s=0}^{1}\sum_{y=0}^{1}\sqrt{\alpha_{s,y}}\iint_{f_{\mathcal{P}_{s,y}}(x,y)>0} f_{\mathcal{P}_{s,y}}(x,y)\mathrm{d}x\mathrm{d}y \tag{57d}$$

$$= 1 - \sum_{s=0}^{1}\sum_{y=0}^{1}\sqrt{\alpha_{s,y}}p_{s,y}. \tag{57e}$$

### E.3 Implementation Details of Our Fairness Certification

We conduct vanilla training and then calculate our certified fairness according to our certification framework. Concretely, in the training step, we train a ReLU network composed of 2 hidden layers of size 20 for 100 epochs with binary cross entropy loss (BCE loss) using an Adam optimizer. The initial learning rate is 0.05 for Crime and German datasets, while for other datasets, the initial learning rate is set 0.001. We reduce the learning rate with a factor of 0.5 on the plateau measured by the loss on the validation set with patience of 5 epochs. In the fairness certification step, we set the region granularity to be 0.005 for certification in the general shifting scenario. We use 90% confidence interval when considering finite sampling error. The codes we used follow the MIT license. All experiments are conducted on a 1080 Ti GPU with 11,178 MB memory.

### E.4 Implementation Details of WRM

The optimization problem of tackling distributional robustness is formulated as:

$$\max_{\mathcal{Q}} \mathbb{E}_{(X,Y)\sim\mathcal{Q}}[\ell(h_\theta(X), Y)] \quad \text{s.t.} \quad \mathrm{dist}(\mathcal{P}, \mathcal{Q}) \le \rho \tag{58}$$

where $\mathrm{dist}(\cdot, \cdot)$ is a predetermined distribution distance metric. Note that the optimization is the same as our certified fairness optimization in Problem 1 except for the fairness constraint.

WRM [43] proposes to use the dual reformulation of the Wasserstein worst-case risk to provide the distributional robustness certificate, which is formulated in the following proposition.

**Proposition 2** ([43], Proposition 1). *Let $\ell : \Theta \times \mathcal{Z} \to \mathbb{R}$ and $c : \Theta \times \mathcal{Z} \to \mathbb{R}_+$ be continuous and let $\phi_\gamma(\theta; z_0) := \sup_{z\in\mathcal{Z}}\{\ell(z; \theta) - \gamma c(z; \theta)\}$. Then, for any distribution $\mathcal{P}$ and any $\rho > 0$,*

$$\sup_{\mathcal{Q}:W_c(\mathcal{P},\mathcal{Q})\le\rho} \mathbb{E}_{\mathcal{Q}}[\ell(\theta; Z)] = \inf_{\gamma\ge 0}\{\gamma\rho + \mathbb{E}_{\mathcal{P}}[\phi_\gamma(\theta; Z)]\} \tag{59}$$

where $W_c(\mathcal{P}, \mathcal{Q}) := \inf_{\pi \in \Pi(\mathcal{P}, \mathcal{Q})} \int_{\mathcal{Z}} c(z, z') d\pi(z, z')$ *is the 1-Wasserstein distance between $\mathcal{P}$ and* $\mathcal{Q}$.

One requirement for the certificate to be tractable is that the surrogate function $\phi_\gamma$ is concave with respect to $Z$, which holds when $\gamma$ is larger than the Lipschitz constant $L$ of the gradient of $\ell$ with respect to $Z$. Since we use the ELU network with JSD loss, we can efficiently calculate $\gamma$ iteratively as shown in Appendix D of [47].

We select Gaussian mixture data for fair comparison. The Gaussian mixture data can be formulated as $P(x|\theta) = \sum_{k=1}^{K} \alpha_k \phi(x|\theta_k)$ where $K$ is the number of Gaussian data, $\alpha_k$ is the proportion of the $k$-th Gaussian, and $\theta_k = (\mu_k, \sigma_k^2)$. In our evaluation, we use 2-dimension Gaussian and mixture data composed of 2 Gaussian ($K = 2$) labeled 0 and 1, respectively. Concretely, we let $\mu_1 = (-2, -0.5), \sigma_1 = 1.0, \alpha_1 = 0.5$ and $\mu_2 = (2, 0.5), \sigma_2 = 1.0, \alpha_2 = 0.5$. The second dimension of input vector is selected as the sensitive attribute $X_s$, and the base rate constraint becomes: $\Pr(Y = 0|X_s < 0) = \Pr(Y = 1|X_s > 0)$. Given the perturbation $\delta \in \mathbb{R}^2$ that induces $X \mapsto X + \delta$, the Wasserstein distance and Hellinger distance can be formulated as follows:

$$W_2(\mathcal{P}, \mathcal{Q}) = \|\delta\|_2, \quad H(\mathcal{P}, \mathcal{Q}) = \sqrt{1 - e^{-\|\delta\|_2^2/8}}. \tag{60}$$

### E.5 More Results of Certified Fairness with Sensitive Shifting and General shifting

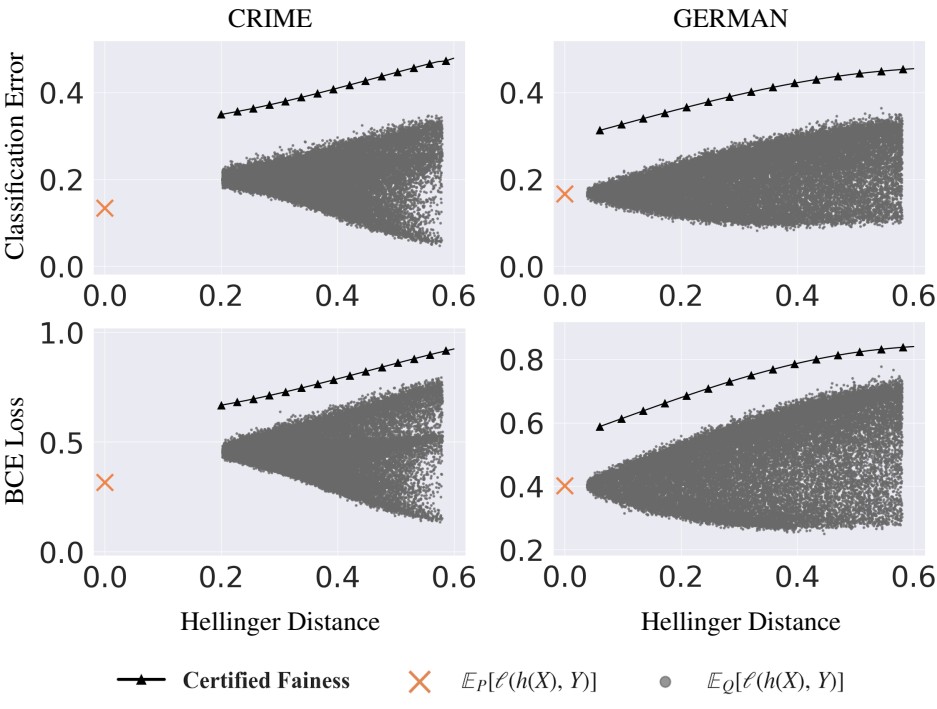

Figure 4: Certified fairness with sensitive shifting on Crime and German.

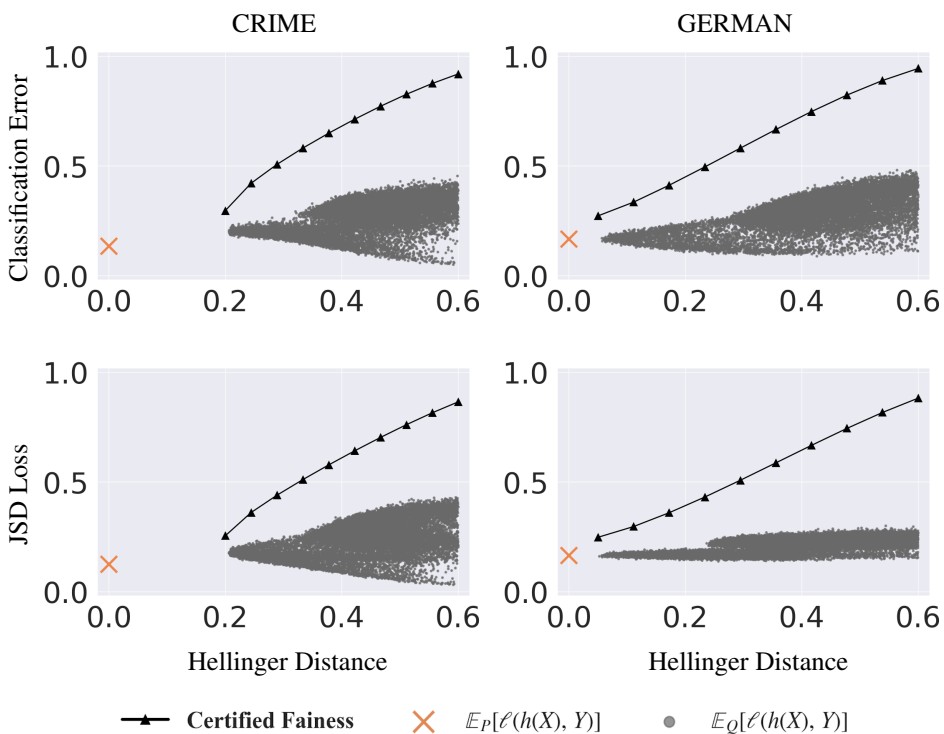

Figure 5: Certified fairness with general shifting on Crime and German.

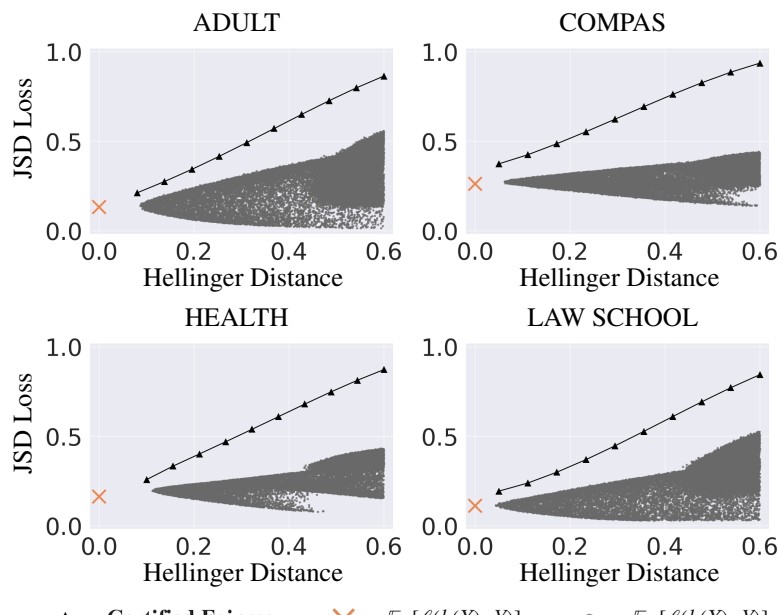

Figure 6: Certified fairness with general shifting using JSD loss on Adult, Compas, Health, and Lawschool.

## E.6 More Results of Certified Fairness with Additional Non-Skewness Constraints

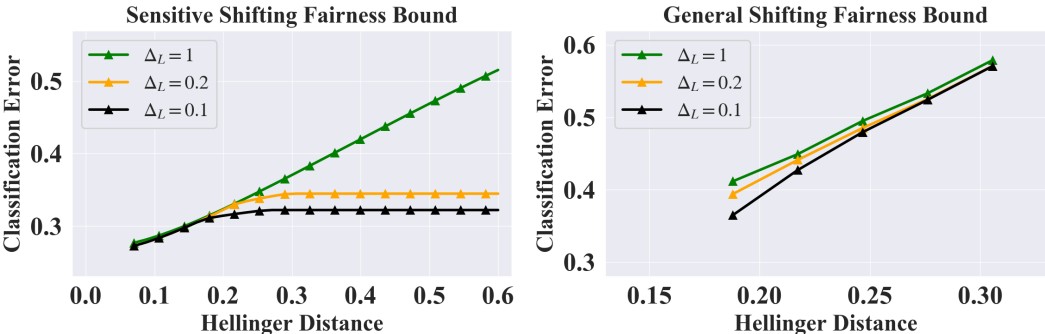

Figure 7: Certified fairness upper bounds with additional non-skewness constraints of labels on Adult. $(|\Pr_{(X,Y)\sim P}[Y=0] - \Pr_{(X,Y)\sim P}[Y=1]| \leq \Delta_L)$

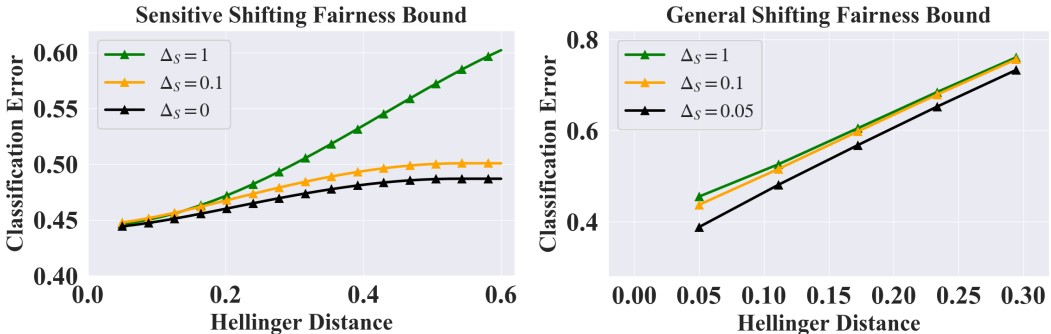

Figure 8: Certified fairness upper bounds with additional non-skewness constraints of sensitive attributes on Compas. $(|\Pr_{(X,Y)\sim P}[X_s=0] - \Pr_{(X,Y)\sim P}[X_s=1]| \leq \Delta_s)$

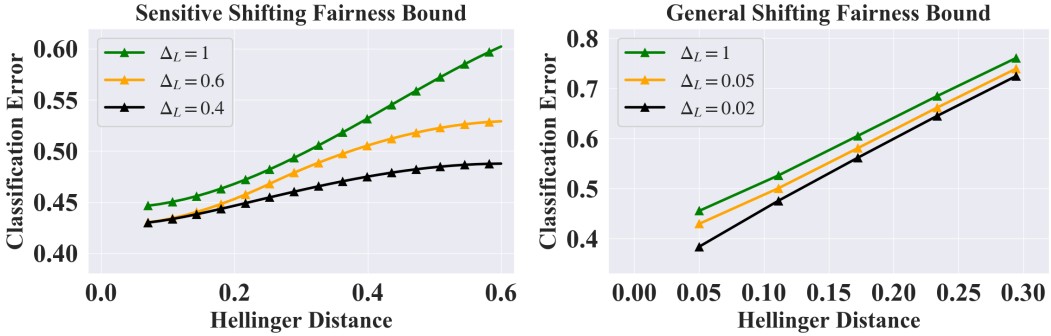

Figure 9: Certified fairness upper bounds with additional non-skewness constraints of labels on Compas. $(|\Pr_{(X,Y)\sim P}[Y=0] - \Pr_{(X,Y)\sim P}[Y=1]| \leq \Delta_L)$

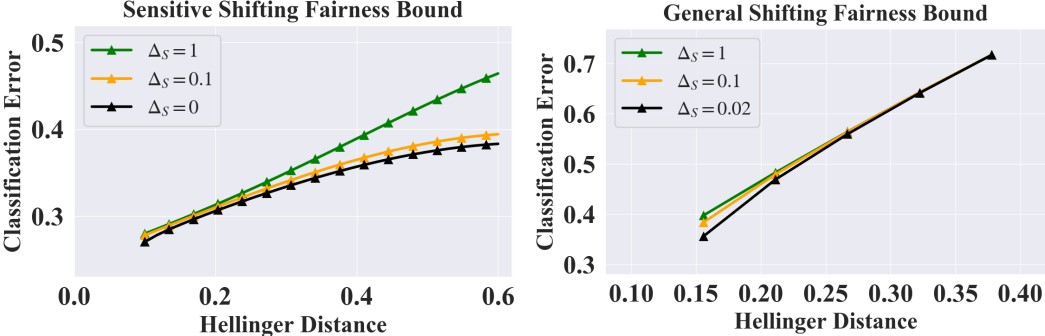

Figure 10: Certified fairness upper bounds with additional non-skewness constraints of sensitive attributes on Health. ($|\Pr_{(X,Y)\sim P}[X_s = 0] - \Pr_{(X,Y)\sim P}[X_s = 1]| \le \Delta_s$)

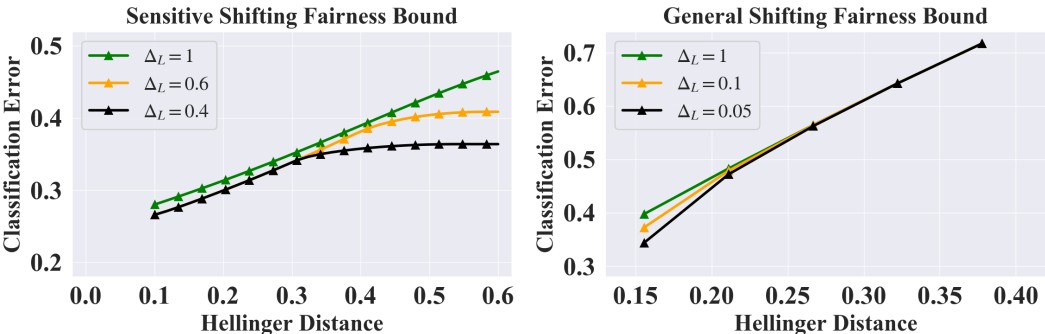

Figure 11: Certified fairness upper bounds with additional non-skewness constraints of labels on Health. ($|\Pr_{(X,Y)\sim P}[Y = 0] - \Pr_{(X,Y)\sim P}[Y = 1]| \le \Delta_L$)

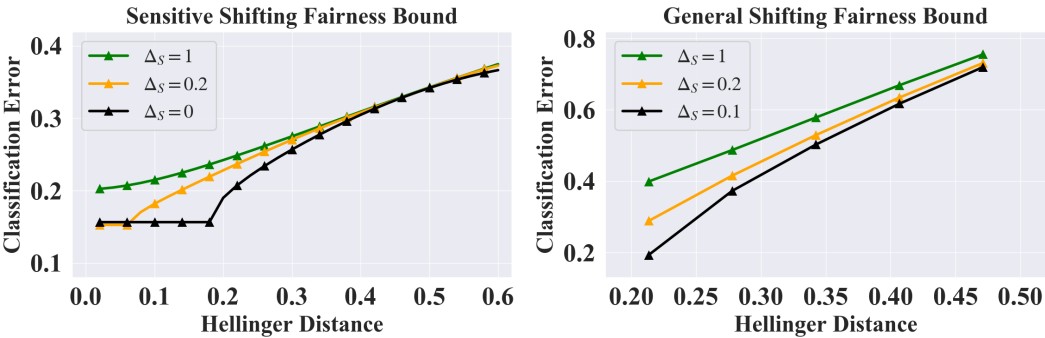

Figure 12: Certified fairness upper bounds with additional non-skewness constraints of sensitive attributes on Lawschool. ($|\Pr_{(X,Y)\sim P}[X_s = 0] - \Pr_{(X,Y)\sim P}[X_s = 1]| \le \Delta_s$)

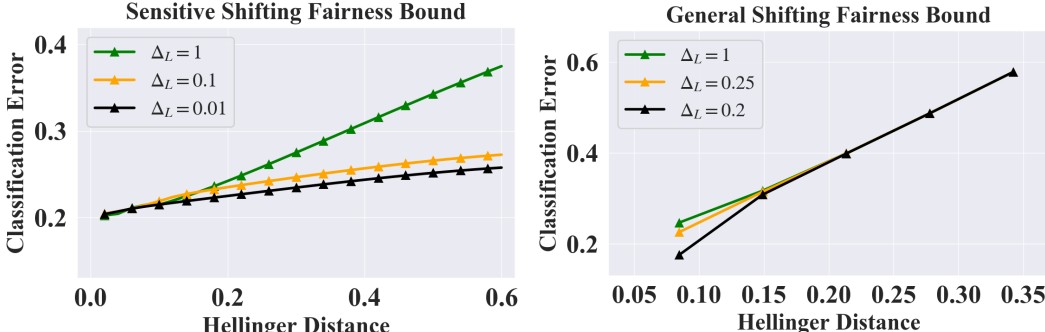

Figure 13: Certified fairness upper bounds with additional non-skewness constraints of labels on Lawschool. $(|\Pr_{(X,Y)\sim P}[Y = 0] - \Pr_{(X,Y)\sim P}[Y = 1]| \leq \Delta_L)$

### E.7 Fair Classifier Achieves High Certified Fairness

We compare the fairness certificate of the vanilla model and the perfectly fair model on Adult dataset to demonstrate that our defined certified fairness in Problem 1 and Problem 2 can indicate the fairness in realistic scenarios. In Adult dataset, we have 14 attributes of a person as input and try to predict whether the income of the person is over 50k \$/year. The sensitive attribute in Adult is selected as the sex. We consider four subpopulations in the scenario: 1) male with salary below 50k, 2) male with salary above 50k, 3) female with salary below 50k, and 4) female with salary above 50k. We take the overall 0-1 error as the loss. The vanilla model is real, and trained with standard training loss on the Adault dataset. The perfectly fair model is hypothetical and simulated by enforcing the loss within each subpopulation to be the same as the vanilla trained classifier's overall expected loss for fair comparison with the vanilla model. From the experiment results in Table 1 and Table 2, we observe that our fairness certificates correlate with the actual fairness level of the model and verify that our certificates can be used as model's fairness indicator: the certified fairness of perfectly fair models are consistently higher than those for the unfair model, for both the general shifting scenario and the sensitive shifting scenario. These findings demonstrate the practicality of our fairness certification.

Table 1: Comparison of the fairness certificate of the vanilla model (an "unfair" model) and the perfectly fair model (a "fair" model) for sensitive shifting. 0-1 error is selected as the loss in the evaluation.

| Hellinger Distance $\rho$ | 0.1 | 0.2 | 0.3 | 0.4 | 0.5 |
|---|---|---|---|---|---|
| Vanilla Model Fairness Certificate | 0.182 | 0.243 | 0.297 | 0.349 | 0.397 |
| Fair Model Fairness Certificate | 0.148 | 0.148 | 0.148 | 0.148 | 0.148 |

Table 2: Comparison of the fairness certificate of the vanilla model (an "unfair" model) and the perfectly fair model (a "fair" model) for general shifting. 0-1 error is selected as the loss in the evaluation.

| Hellinger Distance $\rho$ | 0.1 | 0.2 | 0.3 | 0.4 | 0.5 |
|---|---|---|---|---|---|
| Vanilla Model Fairness Certificate | 0.274 | 0.414 | 0.559 | 0.701 | 0.828 |
| Fair Model Fairness Certificate | 0.266 | 0.407 | 0.553 | 0.695 | 0.824 |