# OpenReview forum: "Certifying Some Distributional Fairness with Subpopulation Decomposition"
_NeurIPS.cc/2022/Conference — NeurIPS 2022 Accept_

### Official Review · Reviewer_iRnk · 2022-07-11

**Rating:** 6
**Confidence:** 4
**Soundness:** 3 good
**Presentation:** 3 good
**Contribution:** 2 fair

**Summary:**

In this paper, the authors propose the certified fairness problem under two kinds of distribution shifts, i.e., the sensitive shifting and the general shifting. They provide the fairness certification framework to deal with the problems. Theoretically, they achieve the upper bound for the two settings and the bound is tight for the sensitive shift setting. Finally, they conduct experiments on six real-world datasets to prove the effectiveness of the method.

**Questions:**

See details in the Strengths and Weaknesses section, especially the points in the Quality, and Significance parts.

**Limitations:**

More limitations are demonstrated in the Strengths and Weaknesses section.


**Strengths And Weaknesses:**

**Originality**

\+ The authors propose a new problem setting, i.e., the certified fairness problem, and develop non-trivial methods to deal with it.

**Quality**

\+ The method provided in this paper is technically sound with detailed proof. The experiments are comprehensive.

\- The relationship between traditional fairness notions (e.g., DP and EO) and the proposed setting should be further discussed. It is well-known that common fairness notions can not be satisfied at the same time in non-trivial settings [1]. As a result, it is not clear why Proposition 1 could guarantee DP and EO simultaneously. More discussions of the tightness of the bounds in Proposition 1 are encouraged.

**Clarity**

\+ The paper is generally well-written. Several small typos exist. For example
 - Line 40: "other other" --> "other"
 - Line 361: "dimensio" --> "dimension"

**Significance**

\- My major concern is about the problem setting proposed in this paper. Fairness problems generally require that several specific statistics match among different subpopulations (e.g., DP and EO) or individuals, which is not the case proposed in this paper. As a result, I am not clear whether the proposed problem setting is practical and sound in real-world scenarios. It would be better if the authors provide real-world examples to demonstrate the soundness of the proposed problem setting.

[1] Kleinberg, Jon, Sendhil Mullainathan, and Manish Raghavan. "Inherent trade-offs in the fair determination of risk scores." arXiv preprint arXiv:1609.05807 (2016).

---

> ### Author Response · Authors · 2022-08-02
> **Author Response [2/2]**
>
> > **Q2**: I am not clear whether the proposed problem setting is practical and sound in real-world scenarios. It would be better if the authors provide real-world examples to demonstrate the soundness of the proposed problem setting.
>
> Thanks for the valuable suggestion. We provide a real-world example in the revised Appendix A with the paragraph title “Example of Fairness Certification”. In short, suppose that the model is deployed in a fair environment, by choosing a proper loss and applying our fairness certification framework, we can guarantee that, say, the crime rate prediction for some particular race group has accuracy larger or equal to some rate. This rate can be used as an indicator of the (worst-case) model fairness in practice. In addition, our derived fairness certification of different training algorithms is also an effective indicator of which algorithm would be fairer during inference in practice.
>
> We further conducted an additional experiment (shown in Appendix F.7) on the Adult dataset. The experiment shows that 1) our fairness certificates correlate with the actual fairness level of the model, and 2) the certified fairness of perfectly fair models are consistently higher than those for the unfair model, for both the general shifting scenario and the sensitive shifting scenario. These findings demonstrate the practicality of our fairness certification.
>
> ------
>
>
> We incorporated all the above responses into the updated version of our paper. Due to the page limit, we put them in Appendix A. If the manuscript is accepted, we will move them to the main text along with more details.

---

> > ### Comment · Reviewer_iRnk · 2022-08-05
> > **Response to rebuttal**
> >
> > Thanks for the authors' response. My concerns about Proposition 1 are well addressed. However, my concerns about the proposed setting remain.
> >
> > In the revised appendix, the athors claim that "the framework provides a rigorous worst-case inaccuracy guarantee for the model’s performance on the fair test distribution. " However, I still wonder if it can guarantee the models' performance on any subgroup. In addition, Proposition 1 requires Problem 2 to return a classifier that achieves certified fairness per group and per label. I am not clear on how to achieve this requirement with a small $\epsilon$.

---

> > > ### Author Response · Authors · 2022-08-05
> > > **Follow-up Response**
> > >
> > > Thank you very much for reading our rebuttal, revision, and prompt reply!
> > >
> > > > **[Q1]**: "The framework provides a rigorous worst-case inaccuracy guarantee for the model’s performance on the fair test distribution. " However, I still wonder if it can guarantee the models' performance on any subgroup.
> > >
> > > Thanks for the question. Note that any population loss $\mathbb{E}[\ell(X,Y)]$ can be plugged into our certification framework. Now, we describe the concrete process for *leveraging our framework to guarantee the model’s performance on any subgroup*: Suppose that there are $S$ groups labeled by $X_s = 1, 2, \dots, S$. To guarantee any subgroup’s worst-case performance, for each group $i$, we let the population loss $\mathbb{E}\_{ (X,Y) \sim Q} [\ell_i(X,Y)] = \mathbb{E}\_{ (X,Y) \sim Q} [\mathbb{1}[h_\theta(X) \neq Y] \cdot \mathbb{1}[X_s = i] / \Pr_{Q}(X_s=i)]$, where $\mathbb{1}[\cdot]$ is the indicator function. Then, our fairness certification framework can compute the upper bound of $\mathbb{E} [\ell_i(X,Y)]$ on the fair test distribution $Q$, which serves as the model’s performance conditioned on subgroup $i$. We run the whole framework for all the $S$ groups respectively, and use the maximum upper bounds of $\mathbb{E} [\ell_i(X,Y)]$ as the model’s performance guarantee on any subgroup.
> > >
> > > Furthermore, we deem that the worst-case inaccuracy guarantee on the fair test distribution can also indicate the worst subgroup performance. When the distance $\rho$ is large enough, the worst-case distribution $Q$ only has support on the subgroup with the largest loss and the worst-case loss equals to the maximal subgroup test loss. With a small $\rho$, the worst-case distribution has the tendency to have large support on the worst subgroup under the distance constraint. The additional experiments in Appendix F.7 also demonstrate that the biased model has a larger certified loss (lower certified fairness) than the perfectly fair model which has the same total loss but performs uniformly on each subgroup, from which we can see that our certified fairness reflects the performance of subgroups and a fair performance across subgroups leads to higher certified fairness.
> > >
> > > We will make this process clear in the revision.
> > >
> > > > **[Q2]**: Proposition 1 requires Problem 2 to return a classifier that achieves certified fairness per group and per label. I am not clear on how to achieve this requirement with a small $\epsilon$.
> > >
> > > Sorry for the confusion. Proposition 1 does not require Problem 2 to return a classifier with absolute certified fairness per group. Instead, Proposition 1 requires Problem 2 to return a classifier that has an upper bound $\epsilon$ for $\bar\ell = \Pr_{(X,Y)\sim Q} [h_\theta(X)\neq Y| Y=y, X_s=i]$ on any *fair distribution*, i.e., we require that: (1) the underlying distribution *(but not the classifier)* is fair; and (2) the classifier has $\epsilon$ loss upper bound. Then, Proposition 1 says, such a classifier, when evaluated on any fair distribution, satisfies $\epsilon$-DP and $\epsilon$-EO. We will make this point clear in the revision and thank you for the comment!

---

> > > > ### Comment · Reviewer_iRnk · 2022-08-06
> > > > **Reply to the follow-up response**
> > > >
> > > > Thanks for your response and I will raise my score to 6. In addition, I strongly recommend the authors discuss more relationships between the proposed setting with subgroup performances in the final version of the paper.

---

> > > > > ### Author Response · Authors · 2022-08-06
> > > > > **Thank you for the valuable suggestion**
> > > > >
> > > > > Thank you very much for your valuable suggestion! We will add concrete discussions about the relationship between the proposed setting with subgroup performances in a separate subsection following the suggestion. Thank you for your questions and suggestions again!

---

> ### Author Response · Authors · 2022-08-02
> **Author Response [1/2]**
>
> We thank the reviewer’s appreciation for our work, and we address the questions below.
>
> > **Q1**: The relationship between traditional fairness notions (e.g., DP and EO) and the proposed setting should be further discussed. It is well-known that common fairness notions can not be satisfied at the same time in non-trivial settings [1]. As a result, it is not clear why Proposition 1 could guarantee DP and EO simultaneously. More discussions of the tightness of the bounds in Proposition 1 are encouraged.
>
> Thanks for the insightful comment. We added the following discussion to Appendix A of the updated paper:
> 1. When $\epsilon = 0$, Proposition 1 can guarantee perfect DP and EO simultaneously. We achieve so because we evaluate with a fair distribution $Q$, where “fair distribution” stands for “equalized base rate” and according to [1, Kleinberg, Mullainathan and Raghavan, 2016] (Theorem 1.1, page 5) both DP and EO are achievable for this fair distribution. This observation in fact motivated us to identify the fair distribution $Q$ for the evaluation since it is this fair distribution that allows the fairness measures to hold at the same time. Therefore, another way to interpret our framework is: given a model, we provide a framework that certifies worst-case “unfairness” bound in the context where multiple fairness goals are simultaneously achievable. Such a worse-case bound serves as the gap to a perfectly fair model and could be a good indicator of the model’s fairness level.
> 2. In practice, $\epsilon$ is not necessarily zero. Therefore, Proposition 1 only provides an upper bound of DP and EO, namely $\epsilon$-DP and $\epsilon$-EO, instead of absolute DP and EO. The approximate fairness guarantee renders our results more general. Meanwhile, there is higher flexibility in simultaneously satisfying approximate fairness metrics (for example when DP = 0, but EO = $\epsilon$, which is plausible for a proper range of $epsilon$, regardless of the distribution $Q$ being fair or not). But again, similar to (1), $\epsilon$-DP and $\epsilon$-EO can be achieved at the same time easily since the test distribution satisfies base rate parity.
>
> Regarding the tightness of the bounds in Proposition 1, we show that the bounds in Proposition 1 are tight. Consider the distribution $Q$ with binary classes and binary sensitive attributes (i.e., $Y$ and $X_s$ are either 0 or 1). When the distribution $Q$ and classifier $h_\theta$ satisfy the conditions that $\mathrm{Pr}\_{Q} [h_\theta(X) \neq Y | Y=0, X_s=0] = \epsilon, \mathrm{Pr}\_{Q} [h_\theta(X) \neq Y | Y=0, X_s=1] = 0$ and $\mathrm{Pr}\_{Q}[Y=0]=1, \mathrm{Pr}\_{Q}[Y=1]=0$, the bounds in Proposition 1 are tight: From $\mathrm{Pr}\_{Q}[Y=0]=1, \mathrm{Pr}\_{Q}[Y=1]=0$, we can observe that $\epsilon$-DP is equivalent to $\epsilon$-EO. From $\mathrm{Pr}\_{Q} [h_\theta(X) \neq Y | Y=0, X_s=0] = \epsilon, \mathrm{Pr}\_{Q} [h_\theta(X) \neq Y | Y=0, X_s=1] = 0$ and $\mathrm{Pr}\_{Q} [h_\theta(X) \neq Y | Y=0, X_s=i] = \mathrm{Pr}\_{Q} [h_\theta(X) = 1 | Y=0, X_s=i]$ for $i = 0$ or $1$, we know that $\epsilon$-EO holds with tightness since $\left|\Pr\_{Q}[h_{\theta}(X) = 1|Y = 0, X_s = 0]-\Pr\_{Q}[h_{\theta}(X) = 1|Y = 0, X_s = 1]\right| = \epsilon$. To this point, we show that both bounds in Proposition 1 are tight.
> We will add related discussion to our revision and thanks for the insightful comment to help improve our work.

---

### Official Review · Reviewer_qSEq · 2022-07-12

**Rating:** 6
**Confidence:** 2
**Soundness:** 2 fair
**Presentation:** 2 fair
**Contribution:** 2 fair

**Summary:**

The authors propose fairness certification on end-to-end model performance based on a fairness constrained distribution.

**Questions:**

- How do you train a classifier on a distribution?
- How do you get $ \rho $ the distance between two distributions in practical datasets?
- How do you differentiate between a fair and a non-fair distribution?

**Limitations:**

In my opinion, there is no negative societal impact of the paper.

**Strengths And Weaknesses:**

Strength:
- The paper seems solid.

Weakness:
- The paper is too dense with notations. An illustrative example (perhaps, earlier in the paper) would improve readability.
- Experimental results such as figures are narrowly explained.

---

> ### Author Response · Authors · 2022-08-02
> **Author Response [2/2]**
>
> > **Q4**: How do you get $\rho$ the distance between two distributions in practical datasets?
>
> Thanks for the comment. Indeed, it is generally challenging to get the precise $\rho$ in practical datasets since we only have finite samples from the dataset. Though $\rho$ estimation based on generative models is feasible (e.g., [b]), such estimation usually lacks a precision guarantee and cannot serve as $\rho$ in our rigorous fairness certificate so we don’t consider them. We will add this discussion in revision. Therefore, to evaluate our fairness certification methods, we compute the fairness certificates under multiple controllable $\rho$’s and compare the upper bound certificates with the actual loss on specially generated real-world distributions where $\rho$ can be precisely computed/known. The results shown in Fig. 1 and Fig. 2 demonstrate the certification tightness of our framework.
>
> *[b] Xu, Yiming, and Diego Klabjan. "Concept drift and covariate shift detection ensemble with lagged labels." 2021 IEEE International Conference on Big Data (Big Data). IEEE, 2021.*
>
> > **Q5**: How do you differentiate between a fair and a non-fair distribution?
>
> As shown in Definition 2, we define a fair distribution if and only if the base rate is equal across all groups conditioned on any label, which indicates that a fair distribution has the property that the probability of being any class is independent of sensitive attribute values following existing fairness literature [c,d]. Our work bounds the model’s worst performance on an arbitrary fair distribution based on the model’s statistics on an accessible but possibly unfair distribution. The base rate fairness of distribution aligns very well with the common fairness definitions in the literature (details in line 110-129 in Section 2).
>
> *[c] Dwork, Cynthia, et al. "Fairness through awareness." Proceedings of the 3rd innovations in theoretical computer science conference. 2012.*
>
> *[d] Zhao, Han, and Geoff Gordon. "Inherent tradeoffs in learning fair representations." Advances in neural information processing systems 32 (2019).*
>
> ------
>
> We incorporated some of the above responses into the updated version of our paper. Due to the page limit, we put them in Appendix A currently. If the manuscript is accepted, we will move them to the main text along with more details.

---

> ### Author Response · Authors · 2022-08-02
> **Author Response [1/2]**
>
> We thank the reviewer’s appreciation for our work, and we address the questions below.
>
> > **Q1**: The paper is too dense with notations. An illustrative example (perhaps, earlier in the paper) would improve readability.
>
> Thanks for the valuable suggestion! We will add an illustrative example of the definition of fairness certification in the main text. In the revised version, we added the following example in Appendix A: The fairness certification problem computes an upper bound of expected loss on any fair test distribution, given the expected loss on a (possibly unfair) training distribution. For example, we can consider the loss to be the inaccuracy within a specific race group in the crime rate prediction task. We assume that the training distribution is possibly unfair with a bounded distribution shift from the test distribution. We can sample and get the inaccuracy under the training distribution. With this inaccuracy, our framework provides a rigorous worst-case inaccuracy guarantee for the model's performance on the *fair test distribution*. Thus, suppose that the model is deployed in a fair environment, we can *guarantee* that the crime rate prediction for some particular race group has an accuracy larger or equal to some rate. This rate can be used as an indicator of the model's fairness.
>
> From the experiment side, we conduct an additional experiment that compares the fairness certificates for a vanilla trained classifier (an “unfair” model) and a perfectly fair classifier (a “fair” model), taking the overall 0-1 error as the loss. The “perfectly fair” classifier is simulated by enforcing the loss within each subpopulation to be the same as the vanilla trained classifier’s overall expected loss. From the experiment results, we observe that our fairness certificates correlate with the actual fairness level of the model and verify that our certificates can be used as the indicator of model fairness: for the fair model, the certified error upper bounds are consistently lower than those for the unfair model, for both general shifting scenario and sensitive shifting scenario. More details of the experiments can be found in Appendix E.7 of the updated paper.
>
> > **Q2**: Experimental results such as figures are narrowly explained.
>
> Thanks for the comment. In the updated version, we added the explanation or reference to the captions of Fig. 1, 2, and 3.
>
> The **goal** of fig. 1 and 2 is to evaluate the tightness of our fairness certification under sensitive shifting and general shifting based on real-world datasets. In particular, in Fig. 1 and Fig. 2, the grey points are the results under generated distributions and the black line is our fairness certificate. From the results, we can see that our certificate bound is tight, especially for sensitive shifting in Fig. 1 with the theoretical tightness guarantee in Thm. 2.
>
> The **goal** of Fig. 3 (a) and (b) is to evaluate our fairness certificate with additional different skewness constraints controlled by $\Delta_s$ corresponding to different lines. The results demonstrate that our certificate framework can flexibly incorporate these constraints and if the added constraints are strict (i.e., smaller $\Delta_s$), the bound is tighter.
>
> Finally, the **goal** of Fig. 3(c) is to compare our certification bound with the distributional robustness bound WRM, which is the SOTA baseline to our best knowledge. WRM uses the dual reformulation of the Wasserstein worst-case risk and the convex surrogate of the optimization objective to provide the distributional robustness certificate, but it is challenging to find the convex surrogate of reformulated fairness constraints to provide a fairness certificate in their framework. The results in Fig. 3(c) suggest that 1) our certified fairness bound is much tighter than WRM given the additional fairness distribution constraint and our optimization framework; 2) with additional fairness constraint, our certificate problem could be infeasible under very small distribution distances since the fairness constrained distribution $Q$ does not exist near the skewed original distribution $P$; 3) with the fairness constraint, we provide non-trivial fairness certification bound even when the distribution shift is large.
>
> > **Q3**: How do you train a classifier on a distribution?
>
> Thanks for the comment. We train the classifier with finite samples from training datasets. In our theoretical results (Appendix E.1), assuming that these finite samples are independently drawn from the underlying distribution, we give high-confidence bounds for the finite sampling error, and the error bounds can be plugged into our distribution-level certificates (Thm. 2 and Thm. 3), to obtain high-confidence certificates on the empirical loss on these finite samples (see Thm. 5 and Thm. 6 in Appendix E.2).

---

### Official Review · Reviewer_f4hd · 2022-07-23

**Rating:** 7
**Confidence:** 3
**Soundness:** 3 good
**Presentation:** 2 fair
**Contribution:** 3 good

**Summary:**

The paper proposes certified fairness as the worst case prediction loss on a distribution Q such that d(P,Q) \leq \rho, where P is the training distribution. The proposed definition is analyzed in two different cases (i) general shifting (ii) sensitive shifting


**Questions:**

Please explain key steps of the proof and add crucial details of experiments to the main paper.

**Limitations:**

yes

**Strengths And Weaknesses:**


S1 The problem of studying fairness under distribution shift is an important topic.

S2 Paper presents interesting ideas to bound loss for shifted distribution.


W1 Theorem 1 is the main result, and it lacks any justification in the main paper. Theorem 2 uses thm 1 and its proof is not that interesting.

W2 Please add more insights for the proof in the main paper and appendix.

W3 Experiments: The process used to generate distribution shift is synthetic and highly specific. Please explain a high level summary of the steps taken and how it impacts the results.

What is the solved used for the optimization problem? How much is the runtime?

Comparison with baselines: Please add a detailed comparison of the proposed approach with baselines and discuss advantages. Fig 3c is the only comparison it does not explain the reasons for difference in quality of different techniques.

Typos:
line 359: following following
line 361: one dimensio

Overall, the paper presents many interesting theoretical ideas but the empirical evidence is weak and presentation can be improved.

---

> ### Author Response · Authors · 2022-08-02
> **Author Response [2/2]**
>
> > **Q4**: What is the solver used for the optimization problem? How much is the runtime?
>
> We use the Sequential Least SQuares Programming (SLSQP) solver implemented in the package of scipy.optimize to solve the optimization problem numerically. The optimization problems in Thm. 2 and Thm. 3 are solved for sensitive shifting and general shifting, respectively, so we individually report the runtime of SLSQP solvers in these two scenarios. The evaluation is done on an Intel core i7-7820x CPU with an NVIDIA 1080 Ti GPU. The results suggest that providing the fairness certificate with our method and the SLSQP solver is quite efficient. Moreover, the fairness certificate for general shifting is more time-consuming than sensitive shifting, which is because: 1) the optimization problem for general shifting in Thm. 3 is more complex than that for sensitive shifting in Thm. 2, leading to a larger runtime per optimization problem; and 2) the grid-based sub-problem construction for general shifting requires solving multiple optimization problems to provide the fairness certificate.
>
> The runtime of fairness certificate with the SLSQP solver on the Adult dataset.
>
> |                                                    | Sensitive shifting | General shifting |
> | -------------------------------------------------- | ------------------ | ---------------- |
> | Runtime Per Optimization Problem Instance (second) | $3.7*10^{-3} $         | $14.2*10^{-3}$       |
> | Total Runtime of One Fairness Certificate (second) | $3.7*10^{-3} $          | $2.11*10^2$              |
>
> Note: In the above table, the “sensitive shifting” column has same runtime per optimization problem and per certificate; while “general shifting” column has much longer per certificate runtime compared to per optimization problem runtime. It is because certifying in “sensitive shifting” scenario only requires solving one optimization problem and certifying in “general shifting” scenario needs to solve multiple problems.
>
> > **Q5**: Comparison with baselines: Please add a detailed comparison of the proposed approach with baselines and discuss advantages. Fig 3c is the only comparison it does not explain the reasons for difference in quality of different techniques.
>
> Thanks for the comment. Since we are the first work providing the certified fairness for an end-to-end learning model, there is no direct baseline to compare with. From the technical perspective, to the best of our knowledge, the only comparable baseline in the literature is WRM (the baseline in Fig 3c), which uses the dual reformulation of the Wasserstein worst-case risk to provide the distributional robustness certificate. WRM observes a convex surrogate of the optimization problem, but it is quite challenging to find a convex surrogate when integrating the non-convex fairness constraint. However, our framework can flexibly deal with the fairness constraint via subpopulation decomposition and solve non-convex optimization problems with grid-based sub-problem construction over the non-convex variables. As a result, we can observe the tighter certification of our method compared with the baseline.
>
> ------
>
> We incorporated the above responses into the updated version of our paper. Due to the page limit, we put them in Appendix A currently. If the manuscript is accepted, we will move them to the main text along with more details.

---

> > ### Comment · Reviewer_f4hd · 2022-08-08
> > **Response to authors**
> >
> > Thanks for answering my questions. I am happy with the responses but I suggest authors add these clarifications to the paper.

---

> > > ### Author Response · Authors · 2022-08-08
> > > **Thank you for the valuable suggestion**
> > >
> > > Thank you very much for the valuable suggestion. We will definitely incorporate these clarifications in our revision. Thank you for helping to improve our work again!

---

> ### Author Response · Authors · 2022-08-02
> **Author Response [1/2]**
>
> We thank the reviewer’s appreciation for our work, and we address the questions below.
>
> > **Q1**: Theorem 1 is the main result, and it lacks any justification in the main paper. Theorem 2 uses thm 1 and its proof is not that interesting.
>
> Thanks for the comment. We added more justification and illustration for Theorem 1 in Appendix A. At a high level, as Appendix D.2 shows, by the unity of probability measure, we can get Eqn. (3): $H(P,Q) \le \rho \iff \sum_{i=1}^N \sqrt{p_i q_i} (1-H^2(P_i,Q_i)) \ge 1-\rho^2$. Based on this relationship, we equalize the constraint between distributions $P$ and $Q$ to subpopulation constraints with regard to $P_i$ and $Q_i$, which brings us to constrained optimization formulation in Thm. 1. Essentially, by decomposing the distributions into subpopulations, we can decompose the overall certified fairness problem (Eqn. (2)) into several constrained optimization subproblems, which is one of our key technics and contributions.
>
> In Thm. 2, the challenge is to deal with the fairness base rate constraint and our core technique is subpopulation decomposition, which transforms the constraints on distribution $P$, $Q$ to those on subpopulation $P_i$, $Q_i$. With the technique, the fairness constraint becomes simple box constraints with regard to subpopulation portions, which enables efficient fairness certification. (added to Appendix A).
>
> > **Q2**: Please add more insights for the proof in the main paper and appendix.
>
> Thanks for the suggestions. In addition to the discussion of Thm 1 and Thm2 above, in Thm. 3, the main challenge is to solve the non-convex optimization problem. Our core technique is the grid-based sub-problem construction over the non-convex variables and the exploitation of convexity of other variables. Concretely, we divide the feasible region regarding non-convex variables into small grids and consider the optimization problem in each region individually. For each sub-problem, we relax the objective by pushing the values of non-convex variables to the boundary in the small region according to the monotonicity and then solve the convex optimization sub-problems (added to Appendix A).
>
> In Thm. 5 and Thm. 6, we consider finite sampling error for Thm. 2 and Thm. 3, respectively. We use Hoeffding’s inequality to bound the finite sampling error of statistics and add the high confidence box constraints to the optimization problems, which can still be proved to be convex (added to Appendix E.3).
>
> > **Q3**: Experiments: The process used to generate distribution shift is synthetic and highly specific. Please explain a high-level summary of the steps taken and how it impacts the results.
>
> Thanks for the suggestion. The distribution generation protocol is detailed in Appendix F.2. We remark that to quantitatively evaluate the fairness certification we need to controllably generate shifted distributions with fairness constraints and precise shift distance $\rho$, and thus we synthesize the shifted distributions from the real-world distributions including Adult, Compas, etc. Existing work adopts similar distribution generation protocols such as [a].
>
> At a high level, we control the portion of each subpopulation (with regard to sensitive attributes and classes) with the base rate fairness constraint. In this way, our protocols can generate distributions with different combinations of subpopulation portions. If the classifier is biased toward one subpopulation (i.e., it achieves high accuracy in the group but low accuracy in others), the worst-case accuracy on generated distribution is low since the portion of the biased subpopulation in the generated distribution can be low; in contrast, a fair classifier which performs uniformly well for each group can achieve high worst-case accuracy (high certified fairness). Therefore, we believe that our protocols can demonstrate real-world training distribution bias as well as reflect the model’s unfairness and certification tightness in real-world scenarios.
>
> *[a] Sinha, Aman, Hongseok Namkoong, and John Duchi. "Certifying Some Distributional Robustness with Principled Adversarial Training." International Conference on Learning Representations. 2018.*

---

### Author Response · Authors · 2022-08-02
**Revision Summary**

We thank all the reviewers for their comments and valuable feedback. We have made the following major updates following the reviews to further improve our work.

1. We add Appendix A, which includes real-world examples that demonstrate the problem formulation and practicality of our fairness certification, according to the suggestions from Reviewer $\textcolor{purple}{qSEq}$ and Reviewer $\textcolor{blue}{iRnk}$.
2. In Appendix A, we discuss the connection between our formulation and classical fairness notions (demographic parity and equalized odds) and the tightness of our bound in Proposition 1, following the suggestion from Reviewer $\textcolor{blue}{iRnk}$.
3. In Appendix A, we include more explanation, illustration, and discussion of the main theoretical results (Theorems 1, 2, and 3), following the suggestion from Reviewer $\textcolor{green}{f4hd}$.
4. In Appendix A, we include more details of the experiments, such as a high-level summary of the distribution generation protocol. The implementation details of our certification (listed in detail response below) will be included in the updated version too.
5. In the captions of Figures 1, 2, and 3, we either summarize the key messages or refer to the sections where key messages are listed to improve the readability.
6. In Appendix F.7, we conduct experiments on the Adult dataset to demonstrate the practicality of our fairness certification, where fair models demonstrate much tighter bounds, i.e., our fairness certification can be used to evaluate and bound the fairness level of real-world models, following the suggestion from Reviewer $\textcolor{purple}{qSEq}$ and Reviewer $\textcolor{blue}{iRnk}$.

All updates are highlighted in blue in our revision. If the manuscript is accepted, all contents in Appendix A will be merged into the main text given the extra page limit for the camera-ready version.

---

### Meta-Review · Area_Chair_KGme · 2022-08-26

**Recommendation:** Accept
**Confidence:** Certain

**Metareview:**

The paper considers an important problem of certifying fairness of trained classifiers based on its performance on some set of fairness constrained distributions. They show the framework for two types of shifts -sensitive shifts and general shifts.

Reviewers are supportive of acceptance. There were many concerns raised by reviewers and reviewers have promised changes and have made them in their revisions. I hope they can incorporate it in their camera ready.

**Award:**

No

---

### Decision · Program_Chairs · 2022-09-14

Accept